

**Shifts in organic sulfur cycling and microbiome composition in the red-tide causing dinoflagellate *Alexandrium minutum* during a simulated marine heat wave**
Elisabeth Deschaseaux[1]*, James O'Brien [1], Nachshon Siboni[1], Katherina Petrou[1,2] and Justin
R. Seymour[1]
[1] University of Technology Sydney, Climate Change Cluster, Ultimo, NSW, 2007, Australia.
[2] University of Technology Sydney, School of Life Sciences, Ultimo, NSW, 2007, Australia.
* **Corresponding author current address:** Dr Elisabeth Deschaseaux, elisabeth.deschaseaux@gmail.com, Centre for Coastal
Biogeochemistry, School of Environment Science and Engineering, Southern Cross University, Lismore, NSW, 2481,
Australia, Ph: (+61) 4 2360 2341.
**Abstract**
The biogenic sulfur compounds dimethylsulfide (DMS), dimethylsulfoniopropionate (DMSP)
and dimethylsulfoxide (DMSO) are produced and transformed by diverse populations of
marine microorganisms and have substantial physiological, ecological and biogeochemical
importance spanning organism to global scales. Understanding the production and
transformation dynamics of these compounds under shifting environmental conditions is
important for predicting their roles in a changing ocean. Here, we report the physiological and
biochemical response of *Alexandrium minutum,* a dinoflagellate with the highest reported
intracellular DMSP content, exposed to a 6 day increase in temperature mimicking coastal
marine heatwave conditions (+ 4°C and + 12°C). Under mild temperature increases (+ 4°C),
*A. minutum* growth was enhanced, with no measurable physiological stress response. However,
under an acute increase in temperature (+ 12°C), *A. minutum* growth declined, photosynthetic
efficiency ($F_V/F_M$) was impaired, and enhanced oxidative stress was observed. These
physiological responses were accompanied by increased DMS and DMSO concentrations
followed by decreased DMSP concentrations. At this higher temperature, we observed a
cascading stress response in *A. minutum*, which was initiated 6h after the start of the experiment
by a spike in DMS and DMSO concentrations and a rapid decrease in $F_V/F_M$. This was followed
by an increase in reactive oxygen species (ROS) and an abrupt decline in DMS and DMSO on
day 2 of the experiment. A subsequent decrease in DMSP coupled with a decline in the growth
rate of both *A. minutum* and its associated total bacterial assemblage coincided with a shift in
the composition of the *A. minutum* microbiome. Specifically, an increase in the relative
abundance of OTUs matching the genus *Oceanicaulis* (17.0%), *Phycisphaeraceae* SM1A02
(8.8%) and *Balneola* (4.9%) as well as a decreased relative abundance of *Maribacter* (24.4%),
*Marinoscillum* (4.7%) and *Seohaeicola* (2.7%), were primarily responsible for differences in



microbiome structure observed between temperature treatments. These shifts in microbiome
structure are likely to have been driven by either the changing physiological state of *A. minutum*
cells, shifts in biogenic sulfur concentrations, or a combination of both. We suggest that these
results point to the significant effect of heatwaves on the physiology, growth and microbiome
composition of the red-tide causing dinoflagellate *A. minutum,* as well as potential implications
for biogenic sulfur cycling processes and marine DMS emissions.
**Keywords**: DMS, DMSP, DMSO, oxidative stress, temperature stress



## 1. Introduction

Many marine phytoplankton produce the organic sulfur dimethylsulfoniopropionate (DMSP) (Zhou et al., 2009;Berdalet et al., 2011;Caruana and Malin, 2014), for which it can function as an antioxidant, osmolyte, grazing deterrent and currency in reciprocal chemical exchanges with heterotrophic bacteria (Stefels, 2000;Sunda et al., 2002;Wolfe et al., 1997;Kiene et al., 2000). Phytoplankton-derived DMSP is in fact a major source of sulfur and carbon for marine heterotrophic bacteria (Kiene et al., 2000), which in turn play a major role in the cycling and turnover of organosulfur compounds in the ocean (Todd et al., 2007;Curson et al., 2011). The subsequent cycling of DMSP into other biogenic sulfur molecules including dimethylsulfide (DMS) and dimethylsulfoxide (DMSO) by a suite of microbial transformation pathways (Kiene et al., 2000;Sunda et al., 2002) and physical drivers (Brimblecombe and Shooter, 1986) have important ecological and biogeochemical implications spanning from cellular to global scales (Sunda et al., 2002;Charlson et al., 1987;DeBose et al., 2008;Van Alstyne et al., 2001;Knight, 2012;Nevitt et al., 1995).

Among DMSP-producing phytoplankton, the dinoflagellate *Alexandrium minutum*, has the highest recorded DMSP cell content, with an average concentration of 14.2 pmol cell$^{-1}$, compared with less than 1 pmol cell$^{-1}$ in most other dinoflagellates (Caruana and Malin, 2014). Blooms of *A. minutum* occur from the Mediterranean Sea to the South Pacific coast in sea surface waters within temperature ranges of 12°C to 25°C (Laabir et al., 2011). Notably, some strains of *Alexandrium,* including *A. minutum,* produce saxitoxins, which lead to paralytic shellfish poisoning (PSP) and are responsible for the most harmful algal blooms in terms of magnitude, distribution and consequences on human health (Anderson et al., 2012).

*A minutum* commonly inhabits shallow coastal and estuarine waters (Anderson, 1998), which are globally experiencing substantial shifts in environmental conditions, including increases in sea surface temperature (SST) associated with climate change (Harley et al., 2006). Although generally less studied than chronic temperature rises associated with global climate change (Frölicher and Laufkötter, 2018), acute temperature increases known as marine heatwaves (MHWs) (Hobday et al., 2016) have recently been demonstrated to be becoming more frequent and persistent as a consequence of climate change (Oliver et al., 2018). Increases in MHW occurrence are anticipated to become particularly frequent within the shallow coastal and




estuarine waters, where *A. minutum* blooms occur (Ummenhofer and Meehl, 2017;Anderson,
85    1998).

Coastal MHW events have recently had dramatic impacts on coastal environments. MHW
events in Western Australian (2011) and the Northeast Pacific (2013-2015) resulted in
significant ecosystem shifts with increases in novel species at the expenses of others (Frölicher
and Laufkötter, 2018). The 2016 MHW that was associated with El Niño Southern Oscillations
resulted in the mass coral bleaching of more than 90% of the Great Barrier Reef (Hughes et al.,
2017). While it is clear that MHWs can have severe consequences on a variety of systems and
organisms, their effects on marine microbes and the biogeochemical processes that they
mediate have rarely been investigated (Joint and Smale, 2017).
While there is evidence that increases in seawater temperature can lead to increased DMSP
and/or DMS concentrations in phytoplankton (McLenon and DiTullio, 2012;Sunda et al.,
2002), it is not clear how a shift in DMSP net production by phytoplankton under acute
temperature stress will alter the composition and function of their associated microbiome and
how, in turn, this will influence biogenic sulfur cycling processes within marine habitats. There
is therefore a pressing need to understand the physiological and biogeochemical consequences
of thermal stress on phytoplankton-bacteria interactions within the context of events such as
MHWs. This is particularly important, given that a shift in the composition of the
phytoplankton microbiome could potentially dictate atmospheric DMS fluxes depending on
whether the bacterial community preferentially cleave or demethylate DMSP (Todd et al.,
2007;Kiene et al., 2000).
The aims of this study were to investigate how acute increases in temperature, such as those
associated with MHW events, alter the physiological state and biogenic sulfur cycling
dynamics of *A. minutum* and determine how these changes might influence the composition of
the *Alexandrium* microbiome. We hypothesized that an abrupt increase in temperature would
lead to physiological impairment (Falk et al., 1996;Robison and Warner, 2006;Iglesias-Prieto
et al., 1992;Rajadurai et al., 2005) and oxidative stress (Lesser, 2006) in *A. minutum*, leading
to an up-regulation of DMSP, DMS and DMSO production (McLenon and DiTullio,
2012;Sunda et al., 2002) in this high DMSP producer, which could ultimately lead to a shift in
the composition of the *A. minutum* microbiome.



## 2. Methods

### 2.1. Culturing and experimental design

Cultures of *Alexandrium minutum* (CS-324), isolated from Southern Australian coastal waters (Port River, Adelaide, 11/11/1988, CSIRO, ANACC's collection) were grown in GSe medium at 18°C and 50 µmol photons $m^{-2}$ $s^{-1}$ under a 12:12 light:dark cycle. One month before the start of each experiment, *A. minutum* cultures were acclimated over four generations to 20°C and 200 µmol photons $m^{-2}$ $s^{-1}$ (14:10 hour light: dark cycle). Cultures were grown to a cell concentration of ~60,000 $mL^{-1}$ before cells were inoculated into fresh GSe medium. Six days prior to the start of experiments, 20 L of GSe medium was inoculated with a cell concentration of 1,140 $mL^{-1}$ (experiment 1, April 2016) and 680 $mL^{-1}$ (experiment 2, June 2016) and aliquots of 500 mL were transferred into 40 individual 750 mL sterile tissue culture flasks. Culture flasks were incubated in four independent water baths (10 flasks in each) and maintained under control conditions of 20°C and 200 µmol photons $m^{-2}$ $s^{-1}$. Temperature and light control was achieved using circulating water heaters (Julabo, country) and programmable LED lights (Hydra FiftyTwo, Company, Country). All cultures were mixed twice daily to keep cells in suspension by gentle swirling.

On Day 1 ($T_0$), five culture flasks from each 20°C water bath were transferred to four new water baths for exposure to experimental treatment temperatures (either 24°C experiment 1; or 32°C, experiment 2), so that each control and experimental water bath contained five flasks. One culture flask from each tank was immediately sampled for baseline measurements of: DMS (2 mL), DMSP and DMSO (1 mL) concentrations, photochemical efficiency (3 mL), algal and bacterial cell counts (1 mL), ROS quantification (1 mL) and DNA extraction (~470 mL). The dissolved DMSP fraction was not determined because preliminary investigations showed that gravity filtration was too time consuming, potentially due to clogging of filters by the large *A. minutum* cells (30 µm diameter), leading to filtration artefacts for DMSP analysis, as have previously been mentioned by Berdalet et al. (2011). At 18:00 on Day 1 ($T_6$), 12:00 on Day 2 ($T_{24}$), 12:00 on Day 5 ($T_{96}$) and 12:00 on Day 6 ($T_{120}$), one flask from each of the eight water baths was removed from the incubation conditions and sampled as described above.

### 2.2. Photosynthetic efficiency measurements

Subsamples for measurement of photosynthetic efficiency were dark adapted for 10 min under aluminium foil and transferred to a quartz cuvette for Pulse Amplitude Modulated (PAM)



fluorometric analysis using a Water PAM (Walz GmbH, Effeltrich, Germany). Once the base
fluorescence ($F_0$) signal had stabilized (measuring light intensity 3, frequency 2s), a saturating
pulse (intensity 12, Width 0.8s) was used to measure the maximum quantum yield ($F_V/F_M$) of
photosystem II (PSII). As base fluorescence is dependent on cell density, the photomultiplier
gain was adjusted and recorded to maintain $F_0$ at a level of 0.2 a.u. before saturating the
photosystem. Samples were kept in suspension during measurements via continuous stirring at
minimal speed inside the quartz cuvette to avoid cells settling.

*2.3.Microalgal and bacterial cell counts*
Subsamples for bacterial cell counts were stained with SYBR Green at a final concentration of
1:10,000 and incubated in the dark for 15 min (Marie et al. 1997). Subsamples for microalgal
cell counts and stained subsamples for bacterial cell counts were diluted 1:10 and 1:100
respectively into sterile GSe medium prior to analysis with a BD Accuri C6 Flow Cytometer
(Becton Dickinson). Phytoplankton cells were discriminated using red auto-fluorescence and
side scatter (SSC), whereas bacterial populations were discriminated and quantified using
SYBR green fluorescence and SSC.

*2.4.Reactive oxygen species measurements*
The presence of reactive oxygen species (ROS) was detected within cultures using the
fluorescent probe 2,7-dichlorodihydrofluorescein-diacetate (CM-H2DCFDA; Molecular
Probes), which binds to ROS and other peroxides (Rastogi et al., 2010). The reagent was
thawed at room temperature for 10 min and activated using 86.5 µL of DMSO, with 5 µL of
activated reagent added to each sample (final concentration 5 µM). Samples were vortexed for
5 sec and incubated at room temperature for 30 min. Samples were then centrifuged at 2,000 g
for 2 min, the supernatant with reagent dye was discarded, and stained cells were resuspended
in 1 mL of PBS, prior to quantification of fluorescence by flow cytometry. Mean green
fluorescence was quantified from cytograms of forward light scatter (FSC) against green
fluorescence. A positive (+ 10 µL of $H_2O_2$) and negative (no ROS added) control of PBS were
run to ensure that detected cell fluorescence was completely attributable to the ROS probe.

*2.5.Sulfur analysis by gas chromatography*
The preparation of all blanks and samples used in the dilution steps described below were
prepared with sterile (0.2 µM filtered and autoclaved) phosphate-buffered saline (PBS, salinity
35ppt) to avoid cell damage from altered osmolarity and to maintain similar physical properties



as seawater during headspace analysis by gas chromatography. Aliquots for DMS analysis were
transferred into 14 mL headspace vials that were immediately capped and crimped using butyl
rubber septa (Sigma Aldrich Pty 27232) and aluminum caps (Sigma Aldrich Pty 27227-U),
respectively. DMSP aliquots were 1:1 diluted with sterile PBS and DMSP was cleaved to DMS
by adding 1 pellet of NaOH to each vial, which was immediately capped and crimped. Samples
were incubated for a minimum of 30 min at room temperature to allow for the alkaline reaction
and equilibration to occur prior to analysis by gas chromatography (Kiene and Slezak, 2006).
At the end of the experiment, alkaline samples used for DMSP analysis were uncapped and left
to vent overnight under a fume hood. On the next day, samples were purged for 10 min with
high purity $N_2$ at an approximate flow rate of 60 mL min$^{-1}$ to remove any remaining DMS
produced from the alkaline treatment. Samples were then neutralized by adding 80 µL of 32 %
HCl and DMSO was converted to DMS by adding 350 µL of 12 % TiCl$_3$ solution to each vial,
which was then immediately capped and crimped (Kiene and Gerard, 1994;Deschaseaux et al.,
2014b). Vials were then heated in a water bath at 50°C for 1h and cooled down to room
temperature prior to analysis by gas chromatography.

DMS, DMSP and DMSO samples were analyzed by 500 µL direct headspace injections using
a Shimadzu Gas Chromatograph (GC-2010 Plus) coupled with a flame photometric detector
(FPD) set at 180°C with instrument grade air and hydrogen flow rates set at 60 mL min$^{-1}$ and
40 mL min$^{-1}$, respectively. DMS was eluted on a capillary column (30 m x 0.32 mm x 5 µm)
set at 120°C using high purity Helium (He) as the carrier gas at a constant flow rate of 5 mL
min$^{-1}$ and a split ratio of five. A six-point calibration curve and PBS blanks were run by 500
µL direct headspace injections prior to subsampling culture flasks using small volumes of
concentrated DMSP.HCl standard solutions (certified reference material WR002, purity 90.3
± 1.8% mass fraction, National Measurement Institute, Sydney, Australia) that were diluted in
sterile PBS to a final volume of 2 mL. A 5-point calibration curve was run prior to DMSO
analysis using DMSO standard solutions (Sigma Aldrich Pty, D2650) diluted in PBS to a final
volume of 2 mL and converted to DMS with TiCl$_3$ in the same manner as the experimental
samples. PBS blanks treated with NaOH and TiCl$_3$ were also run along with the calibration
curves.

216        *2.6. DNA extraction*

Following sub-sampling for the physiological and biogenic sulfur measurements described
above, the remaining 400 mL within each culture flask was filtered onto a 47 mm diameter,




0.22 µm polycarbonate filter (Millipore) with a peristaltic pump at a rate of 80 rpm to retain
cells for DNA analysis. The filters were subsequently stored in cryovials, snap frozen with
liquid nitrogen and stored at -80°C until extraction. DNA extraction was performed using a
bead-beating and chemical lysis based DNA extraction kit (PowerWater DNA Isolation Kit,
MoBio Laboratories) following the manufacturer's instructions. DNA quantity and purity were
checked for each sample using a Nanodrop 2000 (Thermo Fisher Scientific, Wilmington, DE,
USA). Three replicate samples with the highest DNA quantity and purity from the control and
treatment tanks, collected at the beginning ($T_0$) and end ($T_{120}$) of the experiment, were
subsequently sequenced.

*2.7.16S rRNA amplicon sequencing and bioinformatics*
To characterize the bacterial assemblage structure (microbiome) of *A. minutum* cultures, we
employed 16S rRNA amplicon sequencing. We amplified the V1-V3 variable regions of the
16S rRNA gene using the 27F (AGAGTTTGATCMTGGCTCAG, Lane, 1991) and 519R
(GWATTACCGCGGCKGCTG, Turner et al., 1999) primer pairing, with amplicons
subsequently sequenced using the Illumina MiSeq platform (Ramaciotti Centre for Genomics;
Sydney, NSW, Australia) following the manufacturer's guidelines. Raw data files in FASTQ
format were deposited in the National Center for Biotechnology Information (NCBI) Sequence
Read Archive (SRA) under the study accession number PRJNA486692.

Bacterial 16S rRNA gene sequencing reads were analysed using the QIIME pipeline (Caporaso
et al., 2010;Kuczynski et al., 2012). Briefly, paired-end DNA sequences were joined, de novo
Operational Taxonomic Units (OTUs) were defined at 97% sequence identity using UCLUST
(Edgar, 2010) and taxonomy was assigned against the SILVA v128 database (Quast et al.,
2012;Yilmaz et al., 2013). Chimeric sequences were detected using usearch61 (Edgar, 2010)
and together with chloroplast OTUs were filtered from the dataset. Sequences were then
aligned, filtered and rarefied to the same depth to remove the effect of sampling effort upon
analysis.

*2.8.Statistical analysis*
Repeated measures analysis of variance (rmANOVA) models were fitted to the data to quantify
the effects of temperature and time (fixed factors) on all response variables measured in this
experiment (cell density, $F_V/F_M$, ROS, DMS, DMSP and DMSO concentrations) using IBM
SPSS Statistics 20. Assumptions of sphericity were tested using Mauchly's test. In cases where



this assumption was violated, the degrees of freedom were adjusted using the Greenhouse-
Geisser correction factor. Bonferroni adjustments were used for pairwise comparisons. Each
variable was tested for the assumption of normality and log, ln or sqrt transformations were
applied when necessary.

For sequencing data, alpha diversity parameters of the rarefied sequences and Jackknife
Comparison of the weighted sequence data (beta diversity) were calculated in
QIIME (Caporaso et al., 2010). A two-way PERMANOVA with Bray-Curtis similarity
measurements was performed on abundance data of taxonomic groups that contained more
than 1% of total generated OTUs (represent 90.23% of the data) using PAST (Hammer et al.,
2008). In addition, PAST was used to perform non-metric multidimensional scaling (nMDS)
analysis and isolate the environmental parameters (normalised as follows: (x-mean)/stdev) that
contributed the most to the differences between groups using the Bray-Curtis similarity
measure. SIMPER analysis performed with the White *t*-test was used to identify the taxonomic
groups that significantly contributed the most to the shift in bacterial composition in *A.*
*minutum* cultures over time and between temperature treatments.

**3. Results**

*3.1. Algal growth and physiological response*
*A. minutum* cell abundance exponentially increased over time in both the control (20°C) and
24°C temperature treatment, but a significantly faster growth rate ($p = 0.001$, *t*-test) occurred
at 24°C ($2.66 \pm 0.01$ d$^{-1}$; average $\pm$ SE) compared to the 20°C control ($2.57 \pm 0.01$ d$^{-1}$), resulting
in significantly greater cell abundance at 96h ($p = 0.007$) and 120h ($p < 0.001$) (rmANOVA,
**Table 1, Fig. 1a**). On the other hand, the 32°C treatment resulted in decreased growth rates
($2.40 \pm 0.02$ d$^{-1}$ versus $2.58 \pm 0.02$ d$^{-1}$; *t*-test) and significantly lower cell abundance, relative
to the control, at all time points from 6h after the start of the experiment ($p \leq 0.03$; rmANOVA,
**Table 1**, **Fig. 1b**). *A. minutum* abundance demonstrated a marked decline on day 5 in the 32°C
treatment.

No significant difference in the maximum quantum yield ($F_V/F_M$) of *A. minutum* cultures
occurred between 20°C and 24°C until 120h after the start of the experiment, where a
significantly lower $F_V/F_M$ occurred in the 24°C treatment ($p = 0.01$; rmANOVA, **Table 1**, **Fig.**
**2a**). In contrast, $F_V/F_M$ was significantly lower in *A. minutum* cultures maintained at 32°C





compared to the 20℃ at all time points from 6h after the start of the experiment ($p \leq 0.01$)
(rmANOVA, **Table 1**, **Fig. 2b**). However, on days 5 and 6, the $F_V/F_M$ of cultures kept at 32℃
recovered to values (0.72 ± 0.008) close to those of the control (0.75 ± 0.004) (**Fig. 2B**).

*3.2. Reactive oxygen species (ROS)*
Significantly lower concentrations of ROS were measured at 24℃ than at 20℃ at 96h ($p =$
0.003) and 120h ($p = 0.03$) (rmANOVA, **Table 1, Fig. 2c**). In contrast, significantly greater
concentrations of ROS were measured at 32℃ than 20℃ at 24h ($p < 0.001$), 96h ($p = 0.001$)
and 120h ($p = 0.01$) after the start of the experiment (rmANOVA, **Table 1**, **Fig. 2d**). In-line
with the recovery in measured $F_V/F_M$, ROS concentrations in cultures kept at 32℃ started to
decline to values closer to those of the control on days 5 and 6 of the experiment (**Fig. 2d**). A
significant negative correlation between $F_V/F_M$ levels and ROS concentrations was observed
under the 32℃ temperature treatment ($R^2 = 0.623$; $p = 0.02$, $n = 18$; **Fig. 3**).

*3.3. Biogenic sulfur dynamics*
Cellular concentrations of DMSP, DMS and DMSO ranged from 444 ± 33 to 1681 ± 175 fmol
cell$^{-1}$, from 13 ± 1.02 to 87 ± 5 fmol cell$^{-1}$ and from 9 ± 1.41 to 94 ± 24 fmol cell$^{-1}$, respectively,
over both experiments (**Fig. 4**). Concentrations of all three sulfur compounds slowly decreased
over time in all *A. minutum* cultures regardless of the temperature treatment. No significant
difference in DMSP concentration was recorded between 20℃ and 24℃ throughout
experiment 1 ($p > 0.05$; rmANOVA, **Table 1**, **Fig. 4a**), whereas significantly less DMSP was
measured in cells at 32℃ than in the 20℃ control at 96h ($p = 0.02$; rmANOVA, **Table 1**, **Fig.**
**4b**).

Significantly lower DMS concentrations were measured at 24℃ compared to 20℃ at 24h ($p$
$< 0.001$) and 120h ($p = 0.002$) (rmANOVA, **Table 1**, **Fig. 4c**). In contrast, DMS was
significantly higher at 32℃ than 20℃ 6h after the start of the experiment ($p = 0.008$;
rmANOVA, **Table 1**, **Fig. 4d**). A similar pattern was observed for DMSO, where relative to
the controls, it was significantly lower at 24℃ 24h after the start of the experiment ($p = 0.001$;
rmANOVA, **Table 1**, **Fig. 4e**) and significantly greater at 32℃ after 6h and 24h ($p < 0.05$, **Fig.**
**4f**).





### 3.4. Bacterial abundance and composition

Bacterial cell abundance exponentially increased over time at both 20°C and 24°C (**Fig. 5a**). Bacterial abundance was significantly greater at 24°C than at 20°C 120 h after the start of the experiment ($p = 0.05$; rmANOVA, **Table 1**, **Fig. 5a).** However, no significant difference ($p > 0.05$, $t$-test) in bacterial growth rate was observed between 20°C ($4.15 \pm 0.05$ d$^{-1}$) and 24°C ($4.18 \pm 0.01$ d$^{-1}$). In contrast, bacterial growth rate was significantly lower at 32°C than at the 20°C control ($4.05 \pm 0.01$ d$^{-1}$ versus $4.23 \pm 0.02$ d$^{-1}$; $p < 0.001$, $t$-test) (**Fig. 5b**), resulting in significantly lower bacterial cell densities at 24h ($p = 0.002$), 96h ($p = 0.002$) and 120h ($p < 0.001$) relative to the control (rmANOVA, **Table 1**, **Fig. 5b**).

The composition of the initial ($T_0$) *A. minutum* microbiome was consistent across all samples, but then diverged significantly with time and between temperature treatments (**Fig. 6a-b;** Bray-Curtis similarity measurement, Shepard plot stress = 0.0587). A significant temporal shift in bacterial composition occurred at both 20°C and 32°C, with dissimilarities in community composition between $T_0$ and $T_{120}$ of 27% and 42% occurring respectively (SIMPER analysis). Notably, bacterial communities at 32°C differed significantly (two-way PERMANOVA; $p < 0.05$) to 20°C at $T_{120}$, with 32% dissimilarity in community composition. These differences were primarily driven by increased relative abundance of bacterial Operational Taxonomic units (OTUs) within the *Oceanicaulis* (17%), *Phycisphaeraceae SM1A02* (8.8%) and *Balneola* (4.9%) genus along with a decline in the relative abundance of OTUs matching *Maribacter* (24%), *Marinoscillum* (4.7%) and *Seohaeicola* (2.7%) (*Rhodobacter* family) in the 32°C treatment (White test, **Fig. 6c**), with all taxa cumulatively contributing to 63% of the OTU differences between temperature treatments at $T_{120}$ (SIMPER analysis). In the 32°C treatment, differences in microbiome composition between $T_0$ and $T_{120}$ were driven by the elevated levels of ROS, while in the control (20°C) the community shift was principally driven by differences in bacterial and algal cell abundance (**Fig. 6a**; MDS analysis). Similarly, the elevated concentration of ROS as well as the lower FV/FM, lower algal and bacterial cell abundance and lower DMSP, DMS and DMSO concentrations in the 32°C drove the differences in microbiome composition between the temperature treatments (**Fig. 6b**; MDS analysis)

### 4. Discussion

Climate change induced shifts within marine ecosystems are predicted to fundamentally alter the physiology of planktonic organisms and the biogeochemical transformations that they mediate (Finkel et al., 2009;Tortell et al., 2008;Hallegraeff, 2010). Rising seawater



temperatures are one of the major impacts of climate change on marine ecosystems (Harley et
al., 2006), and can be manifested both as long-term gradual increases (IPCC, 2007, 2013) or
intense episodic marine heatwaves (Frölicher and Laufkötter, 2018;Hobday et al., 2016).
Although less examined to date than chronic temperature increases, MHWs are predicted to
become more frequent and severe (Oliver et al., 2018) and have been proposed as a mechanism
for triggering toxic algal blooms (Ummenhofer and Meehl, 2017). Against this back-drop of
changing environmental conditions, microbial production and cycling of dimethylated sulfur
compounds could be particularly relevant because they simultaneously play a role in the stress
response of marine phytoplankton (Berdalet et al., 2011;Deschaseaux et al., 2014a;Sunda et
al., 2002;Wolfe et al., 2002;Stefels and van Leeuwe, 1998) and have been predicted to have
biogeochemical feed-back effects that are relevant for local climatic processes (Charlson et al.,

367 1987).


This study investigated the biogenic sulfur cycling dynamics of *A. minutum,* and its
microbiome, in response to an intense, short-term thermal stress event, akin to the marine heat-
wave events occurring with increasing frequency within coastal habitats (Oliver et al., 2018).
Indeed, MHWs have been defined as an abrupt increase in temperature of at least 3 to 5°C
above climatological average that lasts for at least 3 to 5 days (Hobday et al., 2016). Large
increases in temperature of about 8°C above the yearly average led to red-tides of exceptional
density in San Francisco Bay (Cloern et al., 2005). While a 12°C increase in temperature
constitutes an extreme scenario of MHWs, even for coastal habitats, this experimental
temperature was selected with the intention to induce thermal stress in *A minutum*.

*4.1.Effects of thermal stress on A. minutum growth, physiology and ROS production*

A 4°C increase in temperature resulted in faster algal growth and lower oxidative stress,
indicating that 24°C was close to a temperature optimum for this strain of *Alexandrium*. This
is perhaps not surprising considering that *Alexandrium* species are capable of growing under a
wide range of temperatures from 12°C to 25°C (Laabir et al., 2011). In contrast, a 12°C increase
in temperature resulted in a rapid and clear cascade of physiological responses, indicative of
an acute thermal stress response in *A. minutum*. Overall, *A. minutum* cells exposed to 32°C
immediately exhibited slower growth relative to the 20°C control, suggesting that a 12°C
increase in temperature rapidly led to either an increase in cell death rate or a decrease in cell
division (Rajadurai et al., 2005;Veldhuis et al., 2001). The slower growth rate at 32°C was



coupled with a drop in photosynthetic efficiency and an increase in ROS concentrations, which
are both common stress responses to thermal stress in marine algae (Lesser, 2006;Falk et al.,
1996;Robison and Warner, 2006;Iglesias-Prieto et al., 1992). In fact, these two physiological
responses are often interconnected as increased ROS production generally occurs in both the
chloroplast and mitochondria of marine algae exposed to thermal stress, causing lipid
peroxidation and ultimately leading to a loss in thylakoid membrane integrity (Falk et al., 1996)
and a decrease in the quantum yield of PSII (Lesser, 2006). This was reflected in the positive
correlation observed between the maximum quantum yield of PSII and ROS concentrations.

Although photosynthetic efficiency remained impaired and ROS concentrations remained high
under 32°C until the end the experiment, both biomarkers of stress started to return to values
closer to those of the 20°C control by day 5 and 6 of the experiment. This was most likely at
the expense of a decline in algal abundance since slow growth often coincides with concurrent
cellular repair and photosystem activity recovery (Robison and Warner, 2006). The differential
physiological response between 24°C and 32°C indicates that although cultures of this strain
of *A. minutum* appear to be highly resistant to temperature changes, an abrupt increase in
temperature of 12°C simulating an extreme marine heatwave led to a prolonged (4 day) stress
response. It could also suggest an acclimation period necessary for such an abrupt shift in
temperature, especially since recovery (in $F_V/F_M$ and ROS levels) was observed towards the
end of the experiment.

*4.2.Biogenic sulfur cycling as a response to thermal stress in A. minutum*

Biogenic organic compounds are key compounds in the stress response of phytoplankton, with
evidence they can be used in responses to changes in temperature (Van Rijssel and Gieskes,
2002;Stefels, 2000). An up-regulation of the biogenic sulfur yield was expected as a stress
response to increased temperature in *A. minutum*, through either an increase in cellular DMSP
concentrations, or an increase in DMS via the cleavage of DMSP (McLenon and DiTullio,
2012;Berdalet et al., 2011;Wolfe et al., 2002;Sunda et al., 2002). No significant change in
DMSP concentrations was observed between the control and 24°C treatment, where, as
described above, physiological responses converged to indicate that 24°C was in fact a more
optimal growth temperature for this organism. This temperature optimum was associated with
lower cellular DMS and DMSO concentrations than in the 20°C control. Since algal stress
responses often result in increased cellular sulfur concentrations in dinoflagellates (McLenon



and DiTullio, 2012;Berdalet et al., 2011), it is perhaps not surprising that DMS and DMSO concentrations decreased under what appear to have been more optimal growth temperature conditions.

In contrast to the decreases in DMS and DMSO observed at 24°C, exposure to 32°C resulted in spikes in DMS and DMSO 6h after the start of the experiment, which accompanied decreased algal growth and impaired photosystem II. The increases in DMS and DMSO observed in the 32°C treatment may have resulted from enhanced DMSP exudation from phytoplankton cells during cell lysis (Simó, 2001), resulting in an increasing pool of dissolved DMSP made readily available to bacteria and phytoplankton DMSP-lyases (Riedel et al., 2015;Alcolombri et al., 2015;Todd et al., 2009;Todd et al., 2007). Indeed, although DMSP-lyases can be present both extracellularly and intracellularly in marine bacteria (Yoch et al., 1997), algal DMSP-lyases seem to be exclusively located extra-cellularly (Stefels and Dijkhuizen, 1996), indicating that DMSP cleavage to DMS is mainly possible when DMSP exudes from phytoplankton cells during lysis (Simó, 2001). However, it is notable that lower DMSP concentrations in the 32°C treatment than in the control only occurred on day 4, whereas the spike in DMS and DMSO were evident at the outset of the experiment (6h). Since this decrease in DMSP at 96h was not coupled with an increase in DMS, this could alternatively be indicative of assimilation of DMSP-sulfur by bacterioplankton for *de novo* protein synthesis (Kiene et al., 2000), with this demethylation pathway often accounting for more than 80% of DMSP turnover in marine surface waters. The spike in DMSO measured 6h after the increase in temperature to 32°C most likely indicated rapid DMS oxidation by ROS under thermal stress (Sunda et al., 2002;Niki et al., 2000). At that time however, we found no evidence for ROS build up in *A. minutum* cultures, possibly because ROS concentrations were kept in check by sufficient DMS synthesis and active DMS-mediated ROS scavenging (Lesser, 2006;Sunda et al., 2002). In contrast, 24h after the start of the experiment, increased ROS coincided with an abrupt decline in DMS and DMSO, perhaps suggestive of serial oxidation via active ROS scavenging of both DMS to DMSO and DMSO to methane sulfinic acid (MSNA) (Sunda et al., 2002).

The only previous study that has examined sulfur responses to stress exposure in *A. minutum* examined the effect of physical turbulence by shaking *A. minutum* cultures for up to four days (Berdalet et al., 2011). While the authors of that study also observed slower cell growth as a response to stress exposure, in contrast to our study, cellular DMSP concentrations increased by 20%. Here, a drop in DMSP concentration was observed at 96h between the control and



temperature treatment. Therefore, even though DMSP concentrations were quantified with a similar approach as in Berdalet et al. (2011) (no filtration of the samples with assuming that particulate DMSP concentrations overrule dissolved DMSP and DMS concentrations), it seems that heat stress and turbulence triggered a dissimilar sulfur response to stress in *A. minutum*.

Overall, a 12°C increase in temperature led to lower photosynthetic efficiency, increased oxidative stress and slower cell growth in the red-tide mediating dinoflagellate *A. minutum*. This physiological stress response was coupled with a differential biogenic sulfur cycling as shown by spikes in DMS and DMSO as well as lower DMSP concentrations, most likely translating ROS scavenging and DMSP uptake by bacterioplankton, respectively.

*4.3. A shift in A. minutum associated-bacteria composition triggered by thermal stress*

In light of DMSP and related biogenic sulfur compounds constituting an important source of carbon and sulfur to phytoplankton-associated bacteria (Kiene et al., 2000), it follows that any shift in biogenic sulfur concentrations could influence the microbiome composition of *A. minutum*. Indeed, the most pronounced temporal shift in the composition of the bacterial community associated with *A. minutum* occurred in the 32°C treatment. This shift was primarily characterized by a statistically significant increase in the relative abundance of OTUs classified as members of the *Oceanicaulis, Phycisphaeraceae* and *Balneola* and a significant decrease in OTUs classified as members of the *Maribacter, Marinoscillum* and *Seohaeicola*. To predict any potential role of these key OTUs in biogenic sulfur cycling processes, we screened the genomes of members of these groups using BLAST for four genes commonly involved in DMSP metabolism: *dmdA*, CP000031.2 (Howard et al., 2006); *dddP*, KP639186 (Todd et al., 2009); *tmm*, JN797862 (Chen et al., 2011); and *dsyB*, KT989543 (Kageyama et al., 2018). A BLAST query of the sequences in the NCBI nucleotide collection (nr/nt) database revealed that previously sequenced members of the genera *Maribacter* (taxid:252356, 357 sequences), *Oceanicaulis* (taxid:153232, 36 sequences), *Marinoscillum* (taxid:643701, 23 sequences), *Seohaeicola* (taxid:481178, 18 sequences) and *Balneola* (taxid:455358, 44 sequences) did not possess any homologs of these sulfur cycling genes. While no homologs were found in the genus *SM1A02,* perhaps because very little genomic information is available for this genus. However a close phylogenetic relative to *SM1A02* (99% query cover, 80% identical, E-value = 0.0), and also a member of the *Phycisphaeraceae* family (*P. mikurensis* 10266; genbank accession numbers AP012338.1), possessed significant homologues to all four

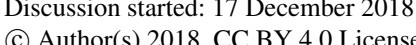



query genes involved in DMSP metabolism: *dmdA* (92% identical, E-value < 0.001), *dddP*
(87% identical, E-value = 0.003), *tmm* (82% identical, E-value = 0.002) and *dsyB* (92%
identical, E-value < 0.001). It is thus possible that the spike in DMS and DMSO concentrations
in the early stage of the 32°C heat treatment was a consequence of (or contributed to) the
preferential recruitment of *Phycisphaeraceae SM1A02*.

Some members of the *Rhodobacter* family such as several members of the *Roseobacter* genus
and *Rhodobacter sphaeroides* are known to possess homologues of either or both *dmdA* and
*ddd* genes, which are responsible for DMSP demethylation and DMSP-to-DMS cleavage,
respectively (Howard et al., 2006;Curson et al., 2008). However, none of the available
reference genomes form *Seohaeicola,* a member of the *Rhodobacteracea,* possessed any
homologs of targeted biogenic sulfur cycling. Similarly, members of the *Maribacter,* which
was the main contributor to the difference in microbiome structure between the control and
thermal stress treatment, are known not to possess DMSP/DMS transformation pathways
(Kessler et al., 2018). Hence, the decline of these taxa in the heat stress treatments, where an
upshift in biogenic sulfur availability occurred, is perhaps not surprising.

Ultimately, the quick conversion of DMSP to DMS (Wolfe et al., 2002) and oxidation of DMS
to DMSO (Sunda et al., 2002) was potentially caused by (or led to) a shift in microbiome
composition towards the preferential growth of sulfur-consuming bacteria (e.g.
*Phycisphaeraceae* SM1A02) at the expense of other types of bacteria (e.g. *Seohaeicola*).
Alternatively, the observed shifts in microbiome structure may have occurred independently to
the biogenic sulfur cycling processes and was instead related to other metabolic shifts in the
heat-stressed *A. minutum*. Notably, the temporal shift in bacterial composition under thermal
stress was associated with increased cellular ROS at the end of the experiment, indicating a
potential link to oxidative stress. However, in light of the phylogenetic patterns discussed
above, this correlation could also reflect a secondary correlation driven by a sulfur-related
cascade response, whereby an increase in ROS could have led to an up-regulation of DMSP
synthesis (McLenon and DiTullio, 2012;Sunda et al., 2002) and DMSP exudation from *A.
minutum* cells (Simó, 2001).



## 5. Conclusion

Abrupt and intense increases in seawater temperatures associated with MHWs are predicted to become more frequent and intense (Oliver et al., 2018) and have the potential to influence the structure of coastal microbial assemblages and the nature of the important biogeochemical processes that they mediate. Here, we hypothesized that an acute increase in temperature, mimicking a coastal MHW, would trigger a physiological and biochemical stress response in the DMSP-producing dinoflagellate *A. minutum*. This response was indeed observed, with evidence for impaired photosynthetic efficiency, oxidative stress, spikes in DMS and DMSO concentrations, a drop in DMSP concentration and a shift in the composition of the *A. minutum* microbiome. These patterns are indicative of a profound shift in the physiological state and biochemical function of an ecologically relevant dinoflagellate under MHW conditions and suggest that MHWs have the potential to not only influence the composition and interactions of coastal microbial food-webs, but re-shape sulfur budgets in coastal waters.

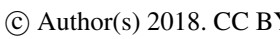


**Acknowledgements**

The work was funded by Australian Research Council grants FT130100218 and DP140101045
to JRS and KP. We thank Rendy Ruvindy and Associate Professor Shauna Murray for
providing the *Alexandrium* cultures and soil extracts. Dr Bonnie Laverock provided advice on
DNA extractions.

**Author contribution**:

ED, KP and JS devised the experimental design. ED and JOB conducted the thermal stress
experiments, including sampling and sample analysis. NS and JOB processed sequencing data
while ED processed the physiological and sulfur data. ED wrote the manuscript with significant
contributions from all co-authors.

**Competing interests**:

The authors declare that they have no conflict of interest.





**Figure captions**

**Figure 1 –** Algal cell abundance in *A. minutum* cultures in experiment 1 (20°C and 24°C) (A) and experiment 2 (20°C and 32°C) (B); average ± SE, $n = 4$.

**Figure 2 –** Photosynthetic efficiency (A, B) and reactive oxygen species (ROS) (C, D) in *A. minutum* cultures in experiment 1 (20°C and 24°C) (A, C) and experiment 2 (20°C and 32°C) (B, D); average ± SE, $n = 4$.

**Figure 3** – Correlation between the photosynthetic efficiency and reactive oxygen species (ROS) in *Alexandrium minutum* under the 32°C thermal stress treatment; $n = 18$.

**Figure 4** – DMSP (A, B), DMS (C, D) and DMSO (E, F) concentrations in *A. minutum* cultures in experiment 1 (20°C and 24°C) (A, C, E) and experiment 2 (20°C and 32°C) (B, D, F); average ± SE, $n = 4$.

**Figure 5 –** Bacterial cell abundance in *A. minutum* cultures in experiment 1 (20°C and 24°C) (A) and experiment 2 (20°C and 32°C) (B); average ± SE, $n = 4$.

**Figure 6** – Multi-dimensional scaling (MDS) of the three phylogenetic groups defined by 16s sequencing of the bacteria population associated with *A. minutum* cultures grown under control conditions (20°C) and acute thermal stress (32°C) at $T_0$ and $T_{120}$ (**A**) and MDS excluding the $T_0$ control (**B**).Vectors represent the factors that most likely drove the shift in bacterial composition between groups. The taxonomic groups that significantly contributed to the difference in bacterial composition between $T_0$ and $T_{120}$ at 32°C [1], between $T_0$ and $T_{120}$ at 20°C [2] and between 32°C and 20°C at $T_{120}$ [3] appear in bold next to the heatmap (**C**), with scaling being based on relative abundance.




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





Table 1. Output of repeated measures analysis of variance (rmANOVA) for algal (CELLS$_A$)

and bacterial (CELLS$_B$) cell abundance, photosynthetic efficiency (F$_V$/F$_M$), oxidative stress

(ROS), dimethylsulfoniopropionate (DMSP), dimethylsulfide (DMS) and dimethylsulfoxide

(DMSO) concentrations as a function of temperature (24°C or 32°C) and time. Numbers in

bold indicate significant data based on the level of significance $p < 0.05$. df1 = numerator df;

df2= denominator df.

| Parameters | | 24°C – mild thermal stress | | | 32°C – mild thermal stress | | |
|---|---|---|---|---|---|---|---|
| | | temperature | time | temperature × time | temperature | time | temperature × time |
| CELLS$_A$ | F | 4.04 | **335** | 4.16 | **27.47** | **237.62** | **8.28** |
| | df1 | 1 | **4** | **4** | **1** | **2.04** | **2.04** |
| | df2 | 6 | **24** | **24** | **6** | **12.26** | **12.26** |
| | p | 0.91 | **< 0.001** | **0.01** | **< 0.001** | **< 0.001** | **0.005** |
| CELLS$_B$ | F | 2.13 | **52.2** | 1.35 | **32.56** | **199.8** | **22.26** |
| | df1 | 1 | **1.29** | 1.29 | **1** | **4** | **4** |
| | df2 | 6 | **7.74** | 7.74 | **6** | **24** | **24** |
| | p | 0.2 | **< 0.001** | 0.3 | **0.001** | **< 0.001** | **< 0.001** |
| F$_V$/F$_M$ | F | **0.42** | **33.43** | **6.90** | **48.79** | **12.58** | **13.11** |
| | df1 | **1** | **4** | **4** | **1** | **1.19** | **1.19** |
| | df2 | **6** | **24** | **24** | **5** | **5.93** | **5.93** |
| | p | **0.54** | **< 0.001** | **0.001** | **0.001** | **0.01** | **0.01** |
| ROS | F | **37.26** | **6.30** | **5.88** | **33.23** | **8.85** | **8.41** |
| | df1 | **1** | **4** | **4** | **1** | **2.32** | **2.32** |
| | df2 | **6** | **24** | **24** | **6** | **13.9** | **13.9** |
| | p | **0.001** | **0.001** | **0.002** | **0.001** | **0.003** | **0.003** |
| DMSP | F | 0.79 | **31.16** | 0.95 | 3.03 | **15.18** | **3.17** |
| | df1 | 1 | **1.56** | 1.56 | 1 | **4** | **4** |
| | df2 | 6 | **9.35** | 9.35 | 6 | **24** | **24** |
| | p | 0.41 | **<0.001** | 0.4 | 0.13 | **< 0.001** | **0.03** |
| DMS | F | **51.5** | **38.73** | 2.01 | 5.08 | **30.77** | **5.23** |
| | df1 | **1** | **2.14** | 2.14 | 1 | **4** | **4** |
| | df2 | **6** | **12.87** | 12.87 | 6 | **24** | **24** |
| | p | **< 0.001** | **< 0.001** | 0.17 | 0.07 | **< 0.001** | **0.004** |
| DMSO | F | **36.56** | **26.64** | **7.21** | 4.68 | **14.74** | **7.14** |
| | df1 | **1** | **4** | **4** | 1 | **4** | **4** |
| | df2 | **6** | **24** | **24** | 6 | **24** | **24** |
| | p | **0.001** | **< 0.001** | **0.001** | 0.07 | **< 0.001** | **0.001** |





**Figure 1**

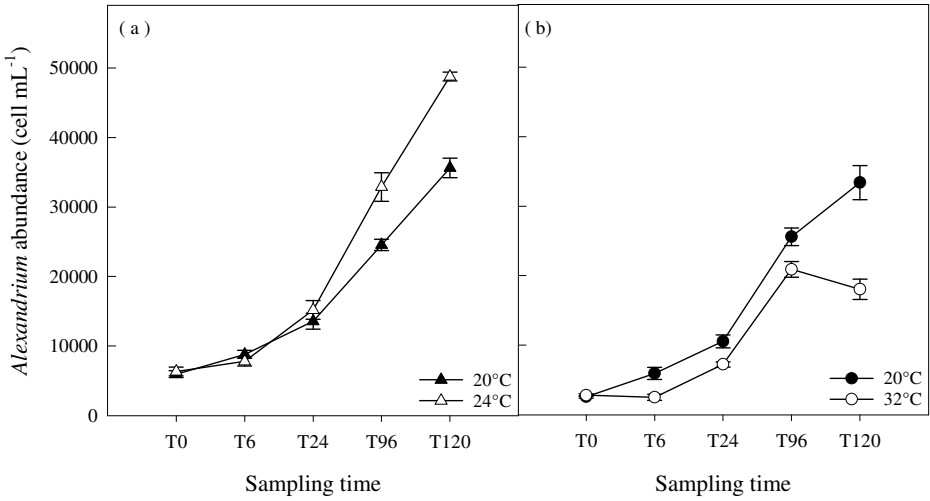



**Figure 2**

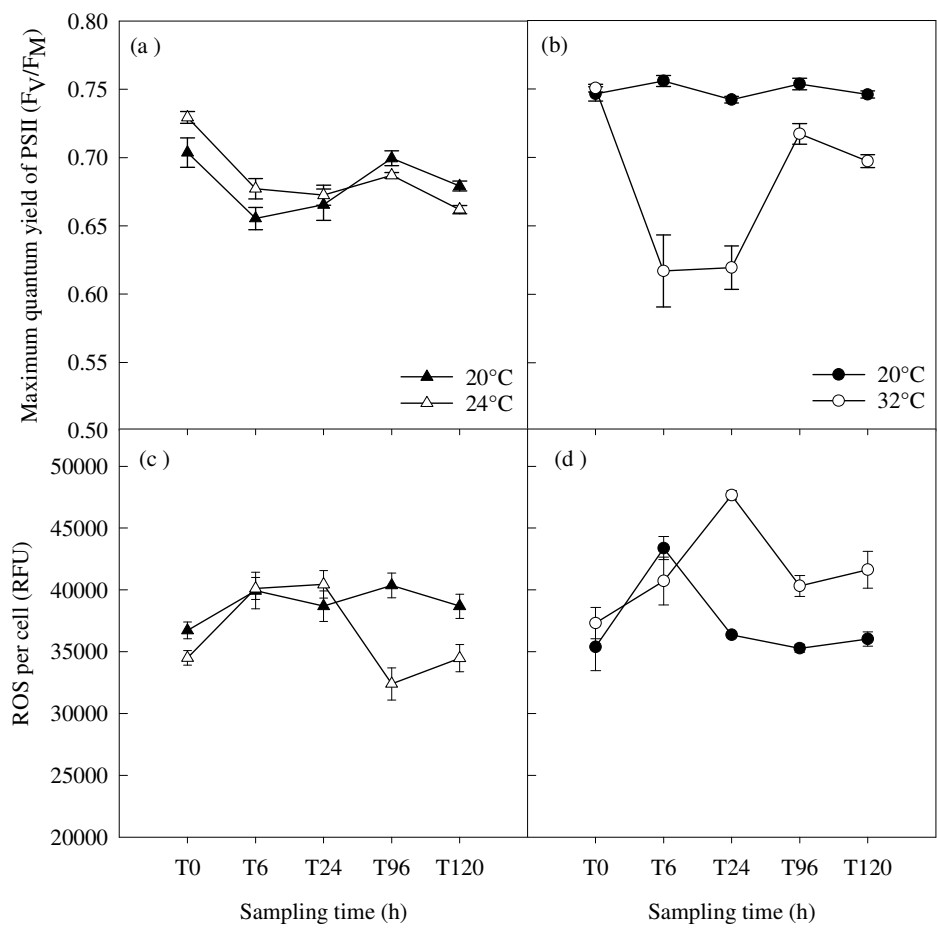





**Figure 3**

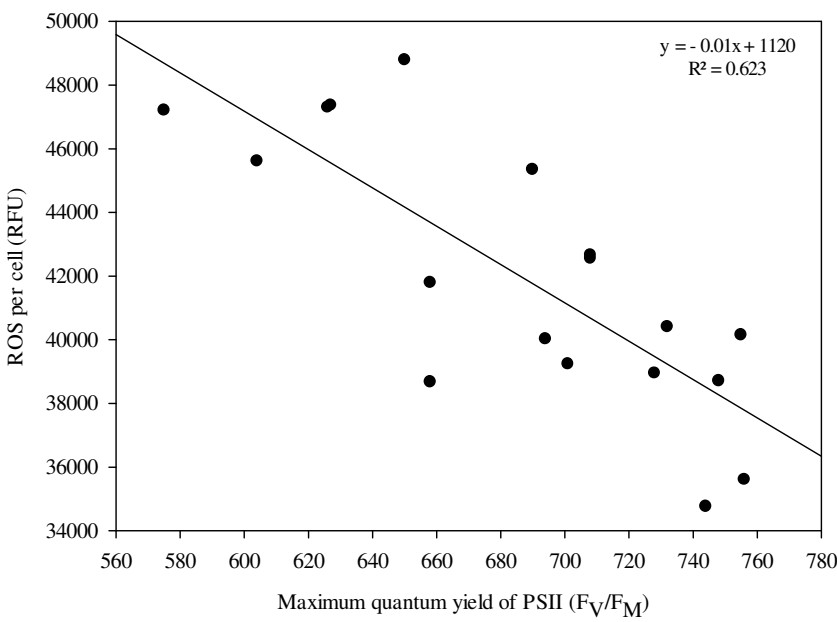



**Figure 4**

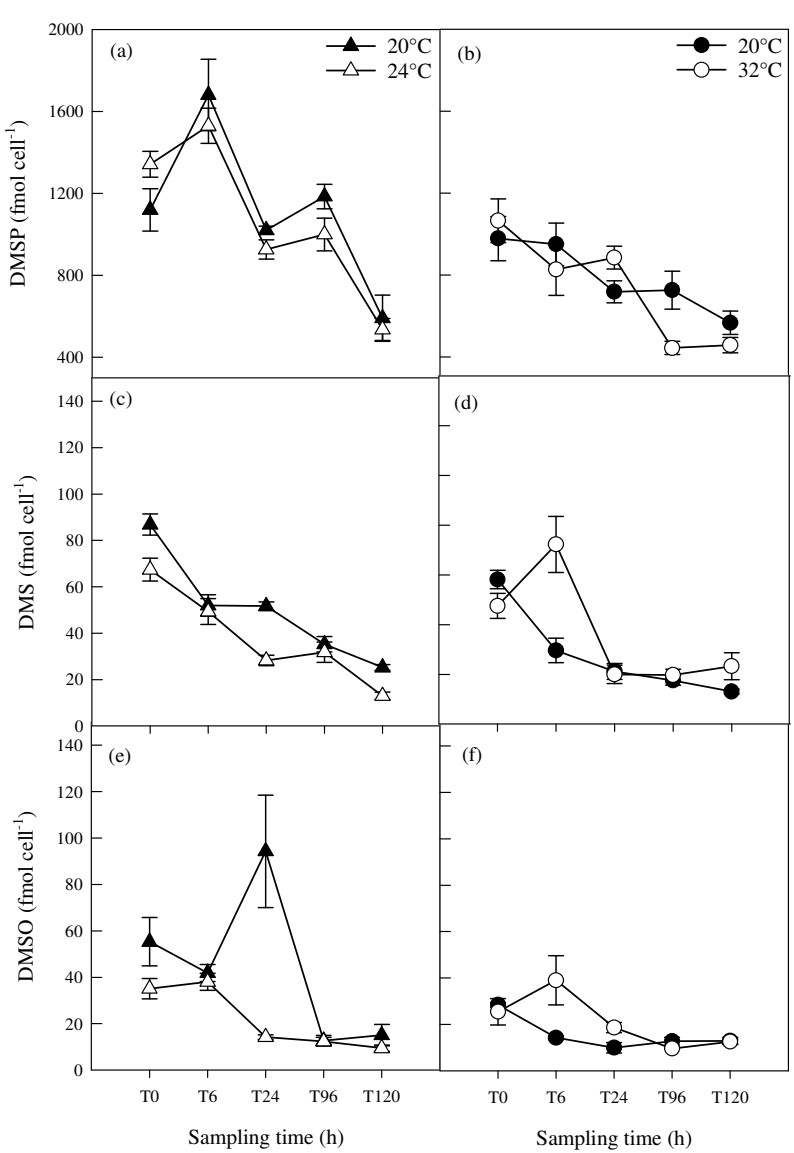



**Figure 5**

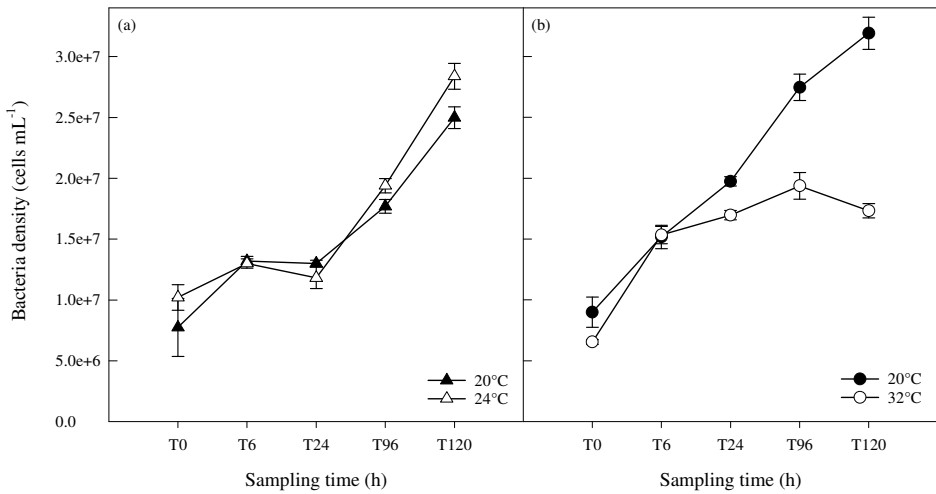





**Figure 6**

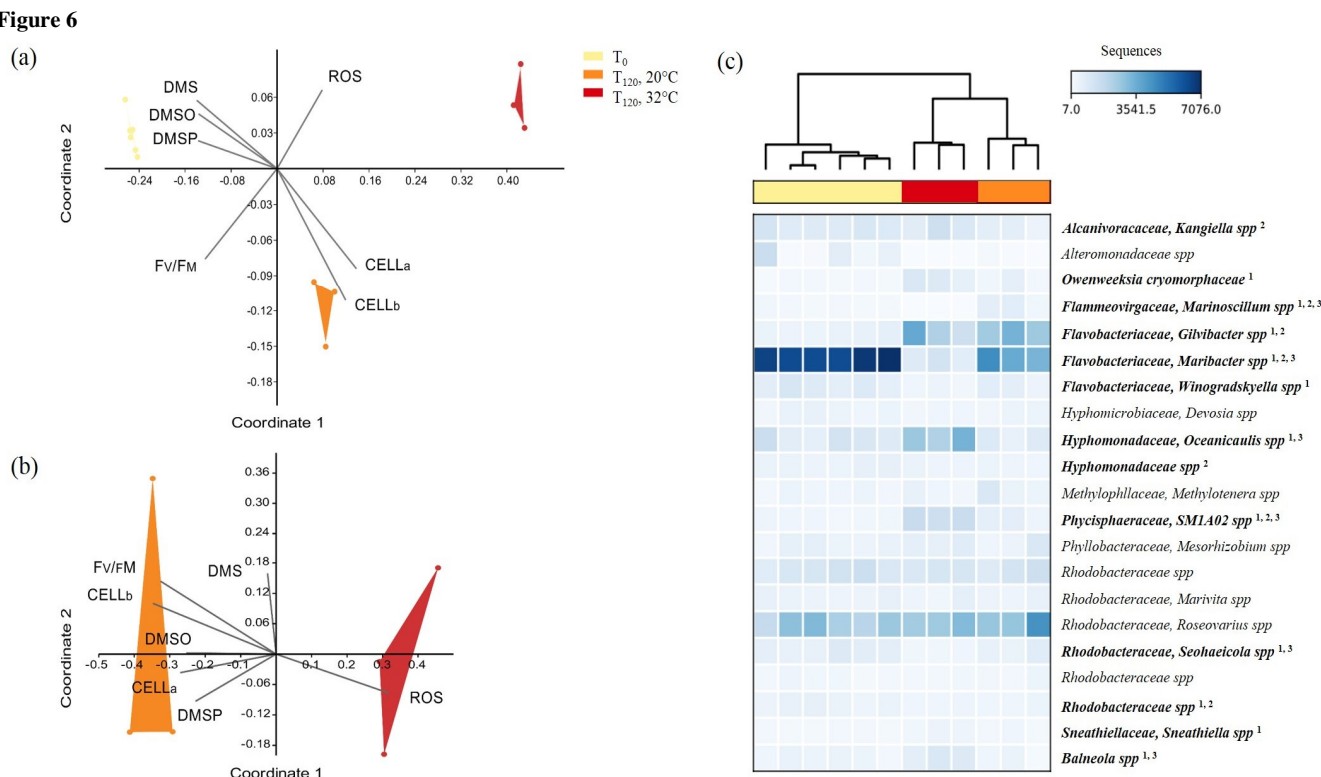