# Peer review of "Elisabeth Deschaseaux1\*, James O'Brien1, Nachshon Siboni1, Katherina Petrou1,2 and Justin"

_Biogeosciences, 2018_

## Referee Comment (RC1) · Anonymous Referee #1 · 5 Jan 2019

General comments

The manuscript by Deschaseaux et al presents a study of how two different levels of temperature change affected 1) the growth and physiological state of the cultured dinoflagellate Alexandrium minutum. 2) the concentrations of the phytoplankton osmolyte DMSP and its degradation products, DMS and DMSO in the cultures, and 3) the taxonomic composition of the bacterial community associated with the cultures, over a six-day period after the temperature shifts. The goal was to assess how temperature increases that might be representative of marine heat waves would affect the phytoplankton and the associated sulfur biogeochemistry and microbial ecology. Marine heat waves are certainly a topic worthy of study, and their effects need investigation.

The authors chose as their control temperature, 20 oC and acclimated the Alexandrium cultures to that temperature before shocking them with +4 and +12 oC increases. The authors don't really justify the choice of their temperatures very well, and their relevance to potential changes in the natural habitats where Alexandrium minutum is found is not evident. The +4 degree temperature shift caused little effects. The +12 degree shift caused effects but what is the environmental relevance of a sudden 12 degree shift? It seems doubtful that a heat wave of that magnitude in a marine system would happen in a short period, if at all. The choice of control temperature of 20 deg was unfortunate. It seems it should have been higher and perhaps the temperature upshift less dramatic. That would have been more realistic.

While there was a clear response of the +12 deg temperature on growth, Fv/Fm and cellular ROS, the effects on DMSP, DMS and DMSO were less clear. There were just a few points with significant differences - not very convincing that it was experimental effect. Most of the discussion is speculation in trying to explain the odd points of higher or lower parameters at particular time points.

In my opinion, the changes in the microbiome were not particularly informative for interpreting the DMS/P/O data. It seems the authors can only speculate on what drove the changes; the MDS analyses are not very convincing for firm conclusions. I know they replicated the treatments in this experiment, but to be really convincing that temperature effects microbiome shifts reproducibly, the entire experiment should be repeated. Also, the bacterial populations would respond to dissolved materials released from the phytoplankton, but there were no measurements aimed at quantifying those releases, making interpretations difficult.

Overall, I feel that the manuscript does not make a substantial contribution as it is, primarily because of the extreme temperature used to produce effects.

Specific comments.

Title. They really didn't study sulfur cycling so I suggest changing the wording.

In Figure 4, the DMSP per cell (0.5 to 1.6 pmol per cell) for Alexandrium minutum is much lower than you report in Introduction for A. minutum (14.2 pmol/cell; line 68). Is there an explanation for that?

L90. When mentioning the 2016 Marine Heat Waves associated with El Nino, give some indication of the temperature increases that occurred.

L131. Julabo, country??

L178. 10 ul of H2O2. Give the concentration of H2O2 added and the final concentration in the sample.

L185. The DMS samples were unfiltered. Were they purged for analysis or did you do static headspace? The static headspace would have a relatively high detection limit. Please provide that value.

L188. From the description, the "DMSP" samples would include DMS that was already in the sample. Was this subtracted from the total DMS after the NaOH?

L192. The transition here to "after the experiment DMSP samples were opened..." is awkward because they didn't describe yet how the DMSP samples were measured. They did this by headspace analysis, which is described further down. I suggest re-organizing to make it clearer. It should be mentioned in methods that all the sulfur compounds were normalized to cell number. But normalizing these parameters to the cells may be misleading. While most of the DMSP will be in the cells, the DMS is most certainly not in the cells. The DMSO has an unknown dissolved and particulate partitioning in their cultures. Referring to them as "cellular" concentrations is not correct.

L225. The description of which samples were sequenced is a little vague. They say they sequenced the three highest DNA samples from each treatment at time zero (so

6 samples) and at T=120 h (6 samples). So, a total of 12 samples were sequenced. Is that correct? By choosing the three samples with the highest DNA could that bias the results?

They filtered 400 ml onto a 0.22 $\mu$m filter, so this would capture both prokaryote and eukaryote DNA. Any interference from all the phytoplankton DNA? They mention removing the chloroplast DNA sequences later on. If the focus here is only the bacteria then the description should be clarified.

L248. I am not an expert in statistics so I can't comment on the approaches used here. But I will say that it wasn't clear to me whether the relative abundance of bacterial groups in each independent replicate was averaged to obtain an error term.

L287 Add word . . .compared to the 20°C CONTROL at all time points. . .

L 289. You say the 32 deg cultures increased to close to those of the control, but were they still significantly lower?

L396. It should be a negative correlation, not positive. L436. The statement that algal DMSP lyases seem to be exclusively extracellular, is not correct. The Stefels paper is the only one that reported extracellular lyase activity, and that study might have methodological issues that led to that conclusion. Evidence against extracellular lyase in Phaeocystis (the same genus studied by Stefels) was presented in del Valle et al (2011, Marine Chemistry, 124: 57-67). Admittedly, few studies have looked at this directly, but even from the bacterial side, most of the evidence from natural water samples (algae and bacteria present) points to intracellular degradation of DMSP. This is based on the fact that an inhibitor of DMSP uptake (e.g. glycine betaine), which does not inhibit DMSP lyases, is nearly 100% effective at blocking DMSP degradation (e.g. Li et al. 2016, Environ. Chem. 13: 266) . If extracellular lyases were important, DMSP degradation would not be blocked by glycine betaine. Furthermore, the bacterial taxa that were identified to have an extracellular lyase (Alcaligenes sp), and its lyase type (dddY), are not prevalent in marine systems (Moran et al Ann Rev Marine Sci, 2012, 4:

523).

L534. In this conclusion section the authors need to make it clear that the effect was with the extreme 12-degree upshift.

Figures 1 and 2. If you are going to connect the data points as a time trend, you should plot them on a linear x-axis rather than a categorical axis, as presently done. The categorical axis gives a misleading impression of the time trend.

Figure 3. The x scale is screwed up. Fv/Fm should be less than 1. It seems they have multiplied it by 100. Please fix.
* * *

---

## Referee Comment (RC2) · Anonymous Referee #2 · 30 Jan 2019

The manuscript reports an experiment where a cultured strain of the dinoflagellate Alexandrium minutum was exposed to temperature increases of 4°C and 12°C. Growth rate, photosynthetic efficiency, oxidative stress, dimethylated sulfur compounds and bacterial community composition were measured over several days. The objective of the experiment was to study if an expected decline in growth rate resulting from impaired physiology was accompanied by up-regulated levels of dimethylated sulfur compounds, and if this matched changes in the microbiome that could be related to sulfur-utilizing bacteria. The environmental context for the lab work is the effects of

marine heat waves on coastal ecosystems, including harmful algal blooms.

Even though the idea behind the experiments is timely and interesting, the experimental conditions chosen generate a little concern, and the actual results are only partially convincing. Perhaps the authors can provide further convincing arguments with the data at hand.

I will give my comments following the order of the manuscript:

L55: The role of DMSP as a grazing deterrent is, at the least, debatable. It is true that the works of Wolfe et al. and Strom et al. suggested deterrence, but more recent work by one of the authors and others (Seymour et al.) indicated DMSP may be more an attractant than a deterrent.

L80: acute temperature increases – should you say also "ephemeral"?

L343-349: I do not like the use of the word "driven" here. Should it be "aligned"? What the MDS analysis shows is that, in the 32°C treatment, differences in the microbiome we aligned with elevated ROS, but that the latter drove the former is just a hypothesis. The same applies to the microbiome composition and abundances in the control, and to the subsequent comparison of variables.

L374: In the case of the San Francisco Bay, MHW were characterized by "increases in temperature of about 8°C above the yearly average". Was it +8°C of the yearly (annual?) average or of the monthly climatological temperatures? +8°C above the annual average would not be too impressive. I mention this because one of my concerns is with the experimental conditions chosen. +12°C seems quite a dramatic treatment. Is there a record of MHW in the S Australian coast where the strain was isolated from? Or perhaps this is not relevant – in any case, what are the temperature shift records of MHW in Australian coasts and elsewhere? More 20°C to 24°C, or 20°C to 32°C?

L396: The correlation is negative, not "positive".

L421-426: I may understand, as a working hypothesis, that optimal growth (hence less

physiological stress) could be associated with lower DMS/P/O concentrations per cell. But it is harder to understand that sulfur concentrations (per culture volume) decreased during the experiment, even with A. minutum being in exponential growth.

L434-438: Why do you say that algal DMSP lyases are exclusively located extra-cellularly? This is definitely not the case in, e.g., Emiliania huxleyi (works by Steinke, Wolfe, Alcolombri).

L446-451: There always is a difficulty when trying to explain and provide experimental evidence for the role of DMS in scavenging ROS: what is first, the decline in DMS or the decline in ROS? It is probably a matter of time scales and potential upregulation by metabolic synthesis. The arguments you provide here carry some assumption that must be explicated.

L492: I would replace DMSP metabolism with DMSP catabolism.

The bacterial community composition characterization was not very informative or illustrative with respect to the cycling of sulfur compounds. Very few of the OTUs that increased their abundances under warming had relatives with genes for sulfur compound transformations. I do not find it any surprising – I think it was too naïve to expect that the bacterial community associated with stressed algae relies mainly on sulfur compounds. Instead, I would expect e.g. opportunistic bacteria. So, I agree with what you say in L513-515. However, I do not agree with your statement in L509-512, at least with the wording used. Quick conversion of DMSP to DMS and oxidation of DMS to DMSO is not a reflection of preferential growth of sulfur-consuming bacteria. Actually, DMSP-to-DMS and DMS-to-DMSO are two processes that do not consume sulfur; if anything, they consume carbon or provide energy. Demethylation of DMSP does lead to sulfur consumption and utilization, and this is a competing process to DMSP cleavage.

Also, you should not base your explanation of the dynamics of the sulfur compounds on the bacterial community alone. There is a potential large role of the dinoflagellate

itself: arrest of methionine synthase activity under growth arrest, DMSP cleavage to DMS by the algal lyases, etc.

From the figures: The (opposite) patterns of ROS and FvFm are pretty consistent. Conversely, the patterns of sulfur compounds are less convincing. The fact that the two controls (20°C) show remarkable differences makes one wonder what would have been the results from repeated perturbations. You may need an extra effort to persuade the readers/reviewers of the robustness of the observed responses with respect to the sulfur compounds.

L531: Only the "very acute" treatment elicited a response.

References: the reference Simó 2001 is repeated.

Figure 4b: The difference between treatments is essentially one time point.

[Figure]

---

## Author Comment (AC1) · 28 Mar 2019

General comments: The manuscript by Deschaseaux et al presents a study of how two different levels of temperature change affected 1) the growth and physiological

state of the cultured dinoflagellate Alexandrium minutum. 2) the concentrations of the phytoplankton osmolyte DMSP and its degradation products, DMS and DMSO in the cultures, and 3) the taxonomic composition of the bacterial community associated with the cultures, over a six-day period after the temperature shifts. The goal was to assess how temperature increases that might be representative of marine heat waves would affect the phytoplankton and the associated sulfur biogeochemistry and microbial ecology. Marine heat waves are certainly a topic worthy of study, and their effects need investigation.

The authors chose as their control temperature, 20 oC and acclimated the Alexandrium cultures to that temperature before shocking them with +4 and +12 oC increases. The authors don't really justify the choice of their temperatures very well, and their relevance to potential changes in the natural habitats where Alexandrium minutum is found is not evident. The +4 degree temperature shift caused little effects. The +12 degree shift caused effects but what is the environmental relevance of a sudden 12 degree shift? It seems doubtful that a heat wave of that magnitude in a marine system would happen in a short period, if at all. The choice of control temperature of 20 deg was unfortunate. It seems it should have been higher and perhaps the temperature upshift less dramatic. That would have been more realistic.

The 20°C control was chosen based on average summer temperatures at the site where this strain of Alexandrium was found (Port River, South Australia). The amplitude of the temperature increase was dictated by preliminary experiments conducted at 20°C, 24°C, 28°C, 30°C and 32°C. The physiology of this strain was found to be highly robust to these temperature increases, with only the 12°C increase in temperature (32°C) leading to a physiological stress response. This may be an adaptation of this strain to shallow coastal environments characterised by dynamic temperature regimes. While a 12 °C increase in temperature might be rare in the environment, this treatment presented an opportunity to investigate the physiological, biochemical and microbial consequences of thermal stress on this relevant phytoplankton in the context of extreme MHWs. We are proposing to provide more details on the choice of temperatures in the manuscript.

While there was a clear response of the +12 deg temperature on growth, Fv/Fm and cellular ROS, the effects on DMSP, DMS and DMSO were less clear. There were just a few points with significant differences - not very convincing that it was experimental effect. Most of the discussion is speculation in trying to explain the odd points of higher or lower parameters at particular time points.

Because DMS(P)(O) turnover in seawater can occur very quickly (Simo et al 2000), it is perhaps not surprising that changes in concentrations occurred only over 1 or 2 time points. However, a clear cascading stress response was still evident with our results, which provides useful information regarding the manner with which biogenic sulfur compounds may play a role in thermal stress tolerance in this relevant dinoflagellate. In response to the Reviewer's concerns, we are proposing to add this information: "Because the turnover of DMS, DMSP and DMSO in biological systems can occur very quickly (Simo et al 2000), DMS and DMSO concentrations can change rapidly, which sometimes makes it difficult to clearly establish cause-effect relationships between physiological stress and the biogenic sulfur response."

In my opinion, the changes in the microbiome were not particularly informative for interpreting the DMS/P/O data. It seems the authors can only speculate on what drove the changes; the MDS analyses are not very convincing for firm conclusions. I know they replicated the treatments in this experiment, but to be really convincing that temperature effects microbiome shifts reproducibly, the entire experiment should be repeated.

We understand the Reviewer's concern, however, the MDS clearly shows a significant difference in the microbiome between the 20°C and 32°C treatments, which corresponded with significant changes in the DMS(P)(O) data. We agree with the Reviewer that the link between the shift in microbiome and DMS(P)(O) concentrations cannot be directly established in this study, and the speculative aspects of the discussion regarding these potential links will be scaled-back. We are proposing to acknowledge that "These shifts in microbiome structure are likely to have been driven by either the changing physiological state of A. minutum cells, shifts in biogenic sulfur concentrations, the presence of other solutes, or a combination of all."

Also, the bacterial populations would respond to dissolved materials released from the phytoplankton, but there were no measurements aimed at quantifying those releases, making interpretations difficult.

The Reviewer makes a fair point that the microbiome composition will be dictated by a range of biochemical factors, and we are offering to acknowledge this point more thoroughly. However, without performing a full metabolomic analysis of the samples, which was beyond the scope and focus of this study, it is not possible to make a priori assessments of the range of chemicals to monitor. Given that A. minutum is a prolific DMSP producer, and it is widely hypothesized that DMSP is a key currency in the chemical exchanges between phytoplankton and bacteria, we focused on the role of Sulphur compounds.

Overall, I feel that the manuscript does not make a substantial contribution as it is, primarily because of the extreme temperature used to produce effects.

The use of laboratory conditions to exactly mimic environmental processes is typically highly challenging from a number of perspectives, and accommodations for environment – laboratory variability often need to be made. Our main goal here was to examine how the heat-stress response of A. minutum was reflected in changes in biogenic sulphur cycling and interactions with the microbiome. The temperature range used here was based on substantial pilot studies (described above) that revealed the large shift in temperature that was required to invoke a physiological stress response in the dinoflagellate species in question. Without increasing the temperature to this level we did not observe a marked physiological response in the dinoflagellate. We propose to more clearly point out the reasons for the choice of temperature used in the study and

feel that our observations provide valuable new insights into how the stress response of dinoflagellates can influence biogenic sulphur cycling in coastal habitats.

Specific comments:

Title. They really didn't study sulfur cycling so I suggest changing the wording.

In response to the authors concerns we are proposing to change the title to "Shifts in dimethylated sulfur concentrations and microbiome composition in the red-tide causing dinoflagellate Alexandrium minutum during a simulated marine heat wave."

In Figure 4, the DMSP per cell (0.5 to 1.6 pmol per cell) for Alexandrium minutum is much lower than you report in Introduction for A. minutum (14.2 pmol/cell; line 68). Is there an explanation for that?

We thank the Reviewer for this comment and propose to state that : "... DMSP concentrations reported in this study were a degree of magnitude lower ($0.42 \pm 0.04$ to $1.63 \pm 1.70$ pmol cell-1) than that previously reported for A. minutum (14.2 pmol cell-1; Caruana and Malin, 2014;Jean et al., 2005). This is potentially because this culture of A. minutum had been isolated from free-living A. minutum for a long time (1988) or because culturing conditions failed to mimic the natural biochemical conditions in which this strain of A. minutum usually grow. This biochemical difference could potentially reflect that this strain of A. minutum in culture is more robust than free-living dinoflagellates of the same species, thereby potentially justifying the need of a $12°C$ increase in temperature to induce thermal-stress."

L90. When mentioning the 2016 Marine Heat Waves associated with El Nino, give some indication of the temperature increases that occurred.

We are proposing to use this information: "The 2016 MHW that was associated with El Niño Southern Oscillations resulted in an $8°C$ increase in sea surface temperature leading to the mass coral bleaching of more than 90% of the Great Barrier Reef (Hughes et al., 2017)".

L131. Julabo, country??

We thank the Reviewer for noticing this omission and will add information has follows: "Temperature and light control was achieved using circulating water heaters (Julabo, USA) and programmable LED lights (Hydra FiftyTwo, AquaIllumination, USA)".

L178. 10 ul of H2O2. Give the concentration of H2O2 added and the final concentration in the sample.

We thank the Reviewer for picking that up and will add information as follows: "A positive (+ 10 $\mu$L of 30% H2O2, final concentration 97mM) and negative (no ROS added) control of PBS were run to ensure that detected cell fluorescence was completely attributable to the ROS probe."

L185. The DMS samples were unfiltered. Were they purged for analysis or did you do static headspace? The static headspace would have a relatively high detection limit. Please provide that value.

Due to the very high DMS concentrations in the Alexandrium cultures, it was possible to analyse DMS concentrations using simple headspace injections as indicated in the methods. The detection limit will be provided as follows: "Detection limit was 50 nM for 500$\mu$L headspace injections"

L188. From the description, the "DMSP" samples would include DMS that was already in the sample. Was this subtracted from the total DMS after the NaOH?

The reviewer makes a good point and we have now corrected our DMSP values to account for the presence of DMS. This information will be included in the methods as follows: "Concentrations obtained in vials treated with NaOH accounted for both DMS and DMSP. Consequently, DMSP concentration in each sample was obtained by subtracting the corresponding DMS concentration." Furthermore, Figure 4 and result section (Biogenic concentrations of DMSP ranged from 424 ± 35 to 1629 ± 170 fmol cell-1) will be amended accordingly. It is important to note that because DMS values

corresponded to less than 5% of DMSP values, this amendment did not lead to any substantial change in the interpretation of our results.

L192. The transition here to "after the experiment DMSP samples were opened:" is awkward because they didn't describe yet how the DMSP samples were measured. They did this by headspace analysis, which is described further down. I suggest reorganizing to make it clearer.

The Reviewer makes a good point and we are offering to reorganize this whole section as follows: "The preparation of all blanks and samples used in the dilution steps described below were prepared with sterile (0.2 $\mu$M filtered and autoclaved) phosphate-buffered saline (PBS, salinity 35ppt) to avoid cell damage from altered osmolarity and to maintain similar physical properties as seawater during headspace analysis by gas chromatography. Aliquots for DMS analysis were transferred into14 mL headspace vials that were immediately capped and crimped using butyl rubber septa (Sigma Aldrich Pty 27232) and aluminum caps (Sigma Aldrich Pty 27227-U), respectively. DMSP aliquots were 1:1 diluted with sterile PBS and DMSP was cleaved to DMS by adding 1 pellet of NaOH to each vial, which was immediately capped and crimped. Samples were incubated for a minimum of 30 min at room temperature to allow for the alkaline reaction and equilibration to occur prior to analysis by gas chromatography (Kiene and Slezak, 2006).

DMS and DMSP samples were analyzed by 500 $\mu$L direct headspace injections using a Shimadzu Gas Chromatograph (GC-2010 Plus) coupled with a flame photometric detector (FPD) set at 180°C with instrument grade air and hydrogen flow rates set at 60 mL min-1 and 40 mL min-1, respectively. DMS was eluted on a capillary column (30 m x 0.32 mm x 5 $\mu$m) set at 120°C using high purity Helium (He) as the carrier gas at a constant flow rate of 5 mL min-1 and a split ratio of five. A six-point calibration curve and PBS blanks were run by 500 $\mu$L direct headspace injections prior to subsampling culture flasks using small volumes of concentrated DMSP.HCl standard solutions (certified reference material WR002, purity 90.3 $\pm$ 1.8% mass fraction, National Measurement Institute, Sydney, Australia) that were diluted in sterile PBS to a final volume of 2 mL. Detection limit was 50 nM for $500\mu$L headspace injections. Concentrations obtained in vials treated with NaOH accounted for both DMS and DMSP. Consequently, DMSP concentration in each sample was obtained by subtracting the corresponding DMS concentration.

Following DMS and DMSP analysis, alkaline samples used for DMSP analysis were uncapped and left to vent overnight under a fume hood. On the next day, samples were purged for 10 min with high purity N2 at an approximate flow rate of 60 mL min-1 to remove any remaining DMS produced from the alkaline treatment. Samples were then neutralized by adding 80 $\mu$L of 32 % HCl and DMSO was converted to DMS by adding 350 $\mu$L of 12 % TiCl3 solution to each vial, which was then immediately capped and crimped (Kiene and Gerard, 1994;Deschaseaux et al., 2014b). Vials were then heated in a water bath at 50°C for 1h and cooled down to room temperature prior to analysis by 500 $\mu$L direct headspace injections on the GC-FPD as described above. A 5-point calibration curve was run prior to DMSO analysis using DMSO standard solutions (Sigma Aldrich Pty, D2650) diluted in PBS to a final volume of 2 mL and converted to DMS with TiCl3 in the same manner as the experimental samples. PBS blanks treated with NaOH and TiCl3 were also run along with the calibration curves. All dimethylated sulfur compounds were normalised to cell density, which best reflects biogenic production."

It should be mentioned in methods that all the sulfur compounds were normalized to cell number.

We thank the Reviewer for pointing out this omission and this details will be added as follows: "All dimethylated sulfur compounds were normalised to cell density, which best reflects biogenic production."

But normalizing these parameters to the cells may be misleading. While most of the DMSP will be in the cells, the DMS is most certainly not in the cells. The DMSO has

an unknown dissolved and particulate partitioning in their cultures. Referring to them as "cellular" concentrations is not correct.

We agree with the Reviewer that DMS and DMSO concentrations should not be referred to as "cellular" since they are most likely not contained within the algal cells. We will thus modify this accordingly throughout the manuscript. However, normalising DMS(P)(O) concentrations to cell numbers remains the most accurate and realistic way to normalise these biogenic sulfur compounds as expressing them in nM without taking algal growth into account would lead to an overestimation of their net production. It is to be noted that DMS and DMSO are commonly expressed per cell (Hatton and Wilson, 2007; Steinke et al., 2011) or per Chl a concentration (Harada et al., 2009; Bucciarelli et al. 2013) in the literature, which is a very similar approach.

L225. The description of which samples were sequenced is a little vague. They sayvthey sequenced the three highest DNA samples from each treatment at time zero (so 6 samples) and at T=120 h (6 samples). So, a total of 12 samples were sequenced. Isvthat correct? By choosing the three samples with the highest DNA could that bias thevresults?

Yes, the Reviewer's interpretation is correct. By using this approach, we had 6 samples at time 0 (all confirmed to have very similar microbial composition), and 6 samples at time 120 (3 from the 24°C and 3 from the 32°C treatment). Samples with the highest DNA quantity (for which DNA extraction was the most successful) were chosen to ensure cost-effective and successful sequencing. However, this approach should not lead to any inherent bias, as the relative abundance of associated microbes should be similar across all replicated samples from the same treatment, regardless of the DNA concentrations. It is also to be noted that the sequence provider normalises the samples according to the DNA concentrations to ensure sufficient reads from all samples.

They filtered 400 ml onto a 0.22 _m filter, so this would capture both prokaryote and eukaryote DNA. Any interference from all the phytoplankton DNA? They mention removing the chloroplast DNA sequences later on. If the focus here is only the bacteria then the description should be clarified.

We used a bacterial specific 16S rRNA primer set, which will specifically target bacterial DNA, so there should be little influence of the eukaryotic DNA in our sequencing results. Chloroplast sequences were indeed removed, further limiting any influence of the eukaryotic DNA.

L248. I am not an expert in statistics so I can't comment on the approaches used here. But I will say that it wasn't clear to me whether the relative abundance of bacterial groups in each independent replicate was averaged to obtain an error term.

We were not entirely sure of what the Reviewer was asking here, but a two-way PERMANOVA with Bray-Curtis takes the response variable of each replicate and the error term is derived from the full data set.

L287 Add word : : :compared to the 20_C CONTROL at all time points:

We thank the Reviewer for noting this omission and will amend this text accordingly.

L 289. You say the 32 deg cultures increased to close to those of the control, but were they still significantly lower?

The Reviewer is correct and we propose to clarify this detail in the manuscript as follows: "However, on days 5 and 6, the FV/FM of cultures kept at 32°C recovered to values (0.72 ± 0.008) close to those of the control (0.75 ± 0.004) (Fig. 2B), although it remained significantly lower than at 20°C (p < 0.01 and p < 0.001 on day 5 and 6, respectively." (lines 302-305)

L396. It should be a negative correlation, not positive.

We thank the Reviewer for noting this typo. This will be changed in the text.

L436. The statement that algal DMSP lyases seem to be exclusively extracellular, is not correct. The Stefels paper is the only one that reported extracellular lyase activity,

and that study might have methodological issues that led to that conclusion. Evidence against extracellular lyase in Phaeocystis (the same genus studied by Stefels) was presented in del Valle et al (2011, Marine Chemistry, 124: 57-67). Admittedly, few studies have looked at this directly, but even from the bacterial side, most of the evidence from natural water samples (algae and bacteria present) points to intracellular degradation of DMSP. This is based on the fact that an inhibitor of DMSP uptake (e.g. glycine betaine), which does not inhibit DMSP lyases, is nearly 100% effective at blocking DMSP degradation (e.g. Li et al. 2016, Environ. Chem. 13: 266). If extracellular lyases were important, DMSP degradation would not be blocked by glycine betaine. Furthermore, the bacterial taxa that were identified to have an extracellular lyase (Alcaligenes sp), and its lyase type (dddY), are not prevalent in marine systems (Moran et al Ann Rev Marine Sci, 2012, 4: 523).

We thank the Reviewer for this comment. We propose to modify this paragraph accordingly: "Although sporadic, the increases in DMS and DMSO observed in the 32°C treatment may have resulted from enhanced intracellular DMSP cleavage by phytoplankton (Del Valle et al., 2011) or enhanced DMSP exudation from phytoplankton cells during cell lysis (SimĬŇ, 2001), resulting in an increasing pool of dissolved DMSP made readily available to both bacteria and phytoplankton DMSP-lyases (Riedel et al., 2015;Alcolombri et al., 2015;Todd et al., 2009;Todd et al., 2007)..."

L534. In this conclusion section the authors need to make it clear that the effect was with the extreme 12-degree upshift.

We thank the Reviewer for this suggestion and propose to make the following change: "Here, we hypothesized that a very acute increase in temperature, mimicking extreme coastal MHWs, would trigger both a physiological and biochemical stress response in the DMSP-producing dinoflagellate A. minutum. This response was indeed observed following a 12°C-increase in temperature, with evidence for impaired photosynthetic efficiency, oxidative stress, spikes in DMS and DMSO concentrations, a drop in DMSP concentration and a shift in the composition of the A. minutum microbiome."

Figures 1 and 2. If you are going to connect the data points as a time trend, you should plot them on a linear x-axis rather than a categorical axis, as presently done. The categorical axis gives a misleading impression of the time trend.

We thank the Reviewer for this suggestion and have modified the x-axis throughout Figures 1, 2, 4 and 5.

Figure 3. The x scale is screwed up. Fv/Fm should be less than 1. It seems they have multiplied it by 100. Please fix.

We thank the Reviewer for noting this. We have now amended Figure 3 accordingly.

Please also note the supplement to this comment:
https://www.biogeosciences-discuss.net/bg-2018-497/bg-2018-497-AC1-supplement.pdf
─────────────────────────────

[Figure]

**Figure 1**

[Figure]

**Fig. 1.**

**Figure 2**

[Figure]

**Fig. 2.**

[Figure]

Figure 3

[Figure]

$$y = -0.01x + 1120$$
$$R^2 = 0.623$$

ROS per cell (RFU)

Maximum quantum yield of PSII ($F_V/F_M$)

**Fig. 3.**

Figure 4

[Figure]

**Fig. 4.**

**Figure 5**

[Figure]

(a) and (b) plots of Bacteria density (cells mL$^{-1}$) vs Sampling time (h). Panel (a) legend: 20℃ (filled triangle), 24℃ (open triangle). Panel (b) legend: 20℃ (filled circle), 32℃ (open circle).

**Fig. 5.**

**Figure 6**

[Figure]

**Fig. 6.**

---

## Author Comment (AC2) · 28 Mar 2019

The manuscript reports an experiment where a cultured strain of the dinoflagellate Alexandrium minutum was exposed to temperature increases of 4_C and 12_C. Growth

rate, photosynthetic efficiency, oxidative stress, dimethylated sulfur compounds and bacterial community composition were measured over several days. The objective of the experiment was to study if an expected decline in growth rate resulting from impaired physiology was accompanied by up-regulated levels of dimethylated sulfur compounds, and if this matched changes in the microbiome that could be related to sulfur-utilizing bacteria. The environmental context for the lab work is the effects of marine heat waves on coastal ecosystems, including harmful algal blooms. Even though the idea behind the experiments is timely and interesting, the experimental conditions chosen generate a little concern, and the actual results are only partially convincing. Perhaps the authors can provide further convincing arguments with the data at hand.

I will give my comments following the order of the manuscript:

L55: The role of DMSP as a grazing deterrent is, at the least, debatable. It is true that the works of Wolfe et al. and Strom et al. suggested deterrence, but more recent work by one of the authors and others (Seymour et al.) indicated DMSP may be more an attractant than a deterrent.

The Reviewer makes a fair point and in fact, it is the cleavage of DMSP to DMS and acrylate that is believed to have strong deterrent properties for grazers, most likely through the presence of acrylate at high concentrations. We propose to change this sentence to read: "Many marine phytoplankton produce the organic sulfur dimethyl sulfoniopropionate (DMSP) (Zhou et al., 2009;Berdalet et al., 2011;Caruana and Malin, 2014), for which it can function as an antioxidant, osmolyte, chemoattractant and currency in reciprocal chemical exchanges with heterotrophic bacteria (Stefels, 2000;Sunda et al., 2002; Kiene et al., 2000;Seymour et al., 2010)."

L80: acute temperature increases – should you say also "ephemeral"?

The Reviewer makes a fair point and we will make this change.

L343-349: I do not like the use of the word "driven" here. Should it be "aligned"? What

the MDS analysis shows is that, in the 32_C treatment, differences in the microbiome we aligned with elevated ROS, but that the latter drove the former is just a hypothesis. The same applies to the microbiome composition and abundances in the control, and to the subsequent comparison of variables.

We agree with the Reviewer's comments and will amend this term accordingly throughout the Results section.

L374: In the case of the San Francisco Bay, MHW were characterized by "increases in temperature of about 8_C above the yearly average". Was it +8_C of the yearly (annual?) average or of the monthly climatological temperatures? +8_C above the annual average would not be too impressive.

The 8°C increase in temperature referred to here was indeed above the monthly average, whereby the MHW occurred during September, with surface water temperatures reaching 22.6°C, while the average temperature for this month is $\sim$ 14°C. We will clarify this statement to read: "Large increases in temperature of about 8°C above the monthly climatological average led to red-tides of exceptional density in San Francisco Bay (Cloern et al., 2005)".

I mention this because one of my concerns is with the experimental conditions chosen. +12_C seems quite a dramatic treatment. Is there a record of MHW in the S Australian coast where the strain was isolated from? Or perhaps this is not relevant – in any case, what are the temperature shift records of MHW in Australian coasts and elsewhere? More 20_C to 24_C, or 20_C to 32_C?

We agree with the Reviewer and in fact, the next sentence of this paragraph acknowledges this point: "While a 12°C increase in temperature constitutes an extreme scenario of MHWs, even for coastal habitats, this experimental temperature was selected with the intention to induce thermal stress in A minutum.". The amplitude of the temperature increase was dictated by preliminary experiments conducted at 20°C, 24°C, 28°C, 30°C and 32°C, with only a 12°C increase in temperature (32°C) leading to a

physiological stress response in this strain of Alexandrium in culture. Although an increase in temperature of this magnitude might be rare in coastal marine systems (which we will acknowledge throughout the manuscript), this experiment presented an opportunity to investigate the biochemical and microbial consequences of thermal stress on this relevant phytoplankton in the context of MHWs.

L396: The correlation is negative, not "positive".

We thank the Reviewer for noting this typo. This will be amended accordingly.

L421-426: I may understand, as a working hypothesis, that optimal growth (hence less physiological stress) could be associated with lower DMS/P/O concentrations per cell. But it is harder to understand that sulfur concentrations (per culture volume) decreased during the experiment, even with A. minutum being in exponential growth.

The Reviewer is correct and we believe that this interpretation is due to our initially unclear description of the data. What we meant was that the DMS(O) concentrations were significantly lower than in the 20°C control, rather than that the concentrations decreased. We will clarify this point as follows: "This temperature optimum was associated with lower DMS and DMSO concentrations than in the 20°C control, although this was only evident 24h after the start of the experiment. Since algal stress responses often result in increased cellular sulfur concentrations in dinoflagellates (McLenon and DiTullio, 2012;Berdalet et al., 2011), it is perhaps not surprising that DMS and DMSO concentrations were lower under what appear to have been more optimal growth temperature conditions."

L434-438: Why do you say that algal DMSP lyases are exclusively located extracellularly? This is definitely not the case in, e.g., Emiliania huxleyi (works by Steinke, Wolfe, Alcolombri).

The Reviewer is correct and we propose to modify our text to reflect this: "Although sporadic, the increases in DMS and DMSO observed in the 32°C treatment may have

resulted from enhanced intracellular DMSP cleavage by phytoplankton (Del Valle et al., 2011) or enhanced DMSP exudation from phytoplankton cells during cell lysis (SimÏŇ, 2001), resulting in an increasing pool of dissolved DMSP made readily available to both bacteria and phytoplankton DMSP-lyases (Riedel et al., 2015;Alcolombri et al., 2015;Todd et al., 2009;Todd et al., 2007)."

L446-451: There always is a difficulty when trying to explain and provide experimental evidence for the role of DMS in scavenging ROS: what is first, the decline in DMS or the decline in ROS? It is probably a matter of time scales and potential upregulation by metabolic synthesis. The arguments you provide here carry some assumption that must be explicated.

The Reviewer makes a good point and we propose to acknowledge the level of uncertainties in this paragraph by saying: "In contrast, 24h after the start of the experiment, increased ROS coincided with an abrupt decline in DMS and DMSO, perhaps suggestive of serial oxidation via active ROS scavenging of both DMS to DMSO and DMSO to methane sulfinic acid (MSNA) (Sunda et al., 2002), although it is always difficult to confidently link DMS(O) and ROS dynamics unless using tracing techniques."

L492: I would replace DMSP metabolism with DMSP catabolism.

The Reviewer makes a fair point and we will amend this terminology accordingly.

The bacterial community composition characterization was not very informative or illustrative with respect to the cycling of sulfur compounds. Very few of the OTUs that increased their abundances under warming had relatives with genes for sulfur compound transformations. I do not find it any surprising – I think it was too naïve to expect that the bacterial community associated with stressed algae relies mainly on sulfur compounds. Instead, I would expect e.g. opportunistic bacteria. So, I agree with what you say in L513-515. However, I do not agree with your statement in L509-512, at least with the wording used. Quick conversion of DMSP to DMS and oxidation of DMS to DMSO is not a reflection of preferential growth of sulfur-consuming bacteria. Actually, DMSP-to-DMS and DMS-to-DMSO are two processes that do not consume sulfur; if anything, they consume carbon or provide energy. Demethylation of DMSP does lead to sulfur consumption and utilization, and this is a competing process to DMSP cleavage.

The Reviewer is correct and we propose to reword this section to clarify our point, which we agree was unclear: "Ultimately, the rapid changes in DMS and DMSO concentrations were potentially caused by (or led to) a shift in microbiome composition towards the preferential growth of sulfur-consuming bacteria (e.g. Phycisphaeraceae SM1A02) at the expense of other types of bacteria (e.g. Seohaeicola). Alternatively, the observed shifts in microbiome structure may have occurred independently to the biogenic sulfur cycling processes and was instead related to other metabolic shifts in the heat-stressed A. minutum. Notably, the temporal shift in bacterial composition under thermal stress was associated with increased cellular ROS at the end of the experiment, indicating a potential link to oxidative stress."

We also propose to acknowledge that: "the change in microbial abundance could have also been triggered by a range of other parameters that were not measured in this study."

Also, you should not base your explanation of the dynamics of the sulfur compounds on the bacterial community alone. There is a potential large role of the dinoflagellate itself: arrest of methionine synthase activity under growth arrest, DMSP cleavage to DMS by the algal lyases, etc.

We agree with the Reviewer and propose to include discussion of these potential processes as follows: "Although sporadic, the increases in DMS and DMSO observed in the 32°C treatment may have resulted from enhanced intracellular DMSP cleavage by phytoplankton (Del Valle et al., 2011) or enhanced DMSP exudation from phytoplankton cells during cell lysis (SimÏŇ, 2001), resulting in an increasing pool of dissolved DMSP made readily available to both bacteria and phytoplankton DMSP-lyases (Riedel et al.,

2015;Alcolombri et al., 2015;Todd et al., 2009;Todd et al., 2007). However, it is notable that lower DMSP concentrations in the 32°C treatment than in the control only occurred on day 4, whereas the spike in DMS and DMSO were evident at the outset of the experiment (6h). Since this decrease in DMSP at 96h was not coupled with an increase in DMS, this could alternatively be indicative of a decrease in methionine synthase activity (McLenon and DiTullio, 2012) or assimilation of DMSP-sulfur by bacterioplankton for de novo protein synthesis (Kiene et al., 2000), with this demethylation pathway often accounting for more than 80% of DMSP turnover in marine surface waters."

From the figures: The (opposite) patterns of ROS and FvFm are pretty consistent. Conversely, the patterns of sulfur compounds are less convincing. The fact that the two controls (20_C) show remarkable differences makes one wonder what would have been the results from repeated perturbations. You may need an extra effort to persuade the readers/reviewers of the robustness of the observed responses with respect to the sulfur compounds.

As described in the method, both experiments were conducted at different times and it was thus not to be excluded that the 2 controls kept at 20'C could present some physiological (Fig. 1 & 2) and biochemical (Fig. 4) differences, which perhaps reflected inherent heterogeneity in biological systems. However, the significant differences that were observed between temperature treatments in each experiment were clearly driven by the increase in temperature since both temperatures (control and experimental) were tested at the same time, on the same culture, and under the exact same experimental conditions of light and GSe medium in each experiment. Because the turnover of DMS(P)(O) in biological systems can occur very quickly (Simo et al 2000), measured changes in DMS(O) concentrations can seem to occur sporadically. However, a clear cascading stress response emerged from these results, which is worth reporting and discussing. We propose to better acknowledged variability and uncertainties in the discussion by saying that: "Because the turnover of DMS, DMSP and DMSO in biological systems can occur very quickly (Simo et al 2000), DMS and DMSO concentrations

can change rapidly, which sometimes makes it difficult to clearly establish cause-effect relationships between physiological stress and the biogenic sulfur response."

L531: Only the "very acute" treatment elicited a response.

We agree with the Reviewer and will amend this sentence as follows: "Here, we hypothesized that a very acute increase in temperature, mimicking extreme coastal MHWs, would trigger both a physiological and biochemical stress response in the DMSP-producing dinoflagellate A. minutum."

References: the reference Simó 2001 is repeated.

We thank the Reviewer for picking this up. This will be amended.

Figure 4b: The difference between treatments is essentially one time point.

We agree with the Reviewer, however, this reflects that differences in sulfur concentration between treatments rely on rapid changes in DMS(O)(P) concentrations, reflective of a quick turnover of DMS(P)(O) in biological systems (Simo et al 2000), which will be better acknowledged in the discussion.

Please also note the supplement to this comment:
https://www.biogeosciences-discuss.net/bg-2018-497/bg-2018-497-AC2-supplement.pdf

Figure 1

[Figure]

**Fig. 1.**

**Figure 2**

[Figure]

**Fig. 2.**

[Figure]

Figure 3

[Figure]

Fig. 3.

[Figure]

**Figure 4**

[Figure]

**Fig. 4.**

**Figure 5**

[Figure]

**Fig. 5.**

**Figure 6**

[Figure]

**Fig. 6.**

---

## Author Response (AR1)

**Responses to Reviewers' comments**

We would like to thank both Reviewers for their comments and suggestions, which have helped improving the wording and reasoning of this manuscript. It is now better acknowledged throughout the manuscript that although a 12°C increase in temperature represents an extreme scenario of Marine Heat Waves (MHWs), this temperature treatment presented an opportunity to investigate the physiological and biochemical response to thermal stress of an ecologically relevant dinoflagellate in the context of MHWs. The choice of temperatures is now more thoroughly justified in the method section. Uncertainties that led to speculative comments are now better acknowledged throughout the manuscript. It is also now better acknowledged that the shifts in microbiome structure at 32°C could be linked to either the physiological and biochemical response of *A. minutum* to thermal stress or by the presence of other solutes that were not measured in this study. Figures have also been modified to reflect the true timeline of this study. We believe that this revised version will now fit with the scope and quality of Biogeosciences and look forward to receiving your feedback.

Responses to Reviewers' comments appear in blue throughout the document.
The manuscript by Deschaseaux et al presents a study of how two different levels of temperature change affected 1) the growth and physiological state of the cultured dinoflagellate Alexandrium minutum. 2) the concentrations of the phytoplankton osmolyte DMSP and its degradation products, DMS and DMSO in the cultures, and 3) the taxonomic composition of the bacterial community associated with the cultures, over a six-day period after the temperature shifts. The goal was to assess how temperature increases that might be representative of marine heat waves would affect the phytoplankton and the associated sulfur biogeochemistry and microbial ecology. Marine heat waves are certainly a topic worthy of study, and their effects need investigation.

The authors chose as their control temperature, 20 oC and acclimated the Alexandrium cultures to that temperature before shocking them with +4 and +12 oC increases. The authors don't really justify the choice of their temperatures very well, and their relevance to potential changes in the natural habitats where Alexandrium minutum is found is not evident. The +4 degree temperature shift caused little effects. The +12 degree shift caused effects but what is the environmental relevance of a sudden 12 degree shift? It seems doubtful that a heat wave of that magnitude in a marine system would happen in a short period, if at all. The choice of control temperature of 20 deg was unfortunate. It seems it should have been higher and perhaps the temperature upshift less dramatic. That would have been more realistic.

The 20°C control was chosen based on average summer temperatures at the site where this strain of *Alexandrium* was found (Port River, South Australia). The amplitude of the temperature increase was dictated by preliminary experiments conducted at 20°C, 24°C, 28°C, 30°C and 32°C. The physiology of this strain was found to be highly robust to these temperature increases, with only the 12°C increase in temperature (32°C) leading to a physiological stress response. This may be an adaptation of this strain to shallow coastal environments characterised by dynamic temperature regimes. While a 12 °C increase in temperature might be rare in the environment (which we now acknowledge more clearly on **lines 143-146**), this treatment presented an opportunity to investigate the physiological, biochemical and microbial consequences of thermal stress on this relevant phytoplankton in the context of extreme MHWs. We have now aimed to better justify this aspect of the study **on lines 141-143 and 390-393, 404-406**.

While there was a clear response of the +12 deg temperature on growth, Fv/Fm and cellular ROS, the effects on DMSP, DMS and DMSO were less clear. There were just a few points with significant differences - not very convincing that it was experimental effect. Most of the discussion is speculation in trying to explain the odd points of higher or lower parameters at particular time points.

Because DMS(P)(O) turnover in seawater can occur very quickly (Simo et al 2000), it is perhaps not surprising that changes in concentrations occurred only over 1 or 2 time points. However, a clear cascading stress response was still evident with our results, which provides useful information regarding the manner with which biogenic sulfur compounds may play a role in thermal stress tolerance in this relevant dinoflagellate. However, in response to the Reviewer's concerns, this is now better acknowledged in the discussion (**see lines 494-498**).

In my opinion, the changes in the microbiome were not particularly informative for interpreting the DMS/P/O data. It seems the authors can only speculate on what drove the changes; the MDS analyses are not very convincing for firm conclusions. I know they replicated the treatments in this experiment, but to be really convincing that temperature effects microbiome shifts reproducibly, the entire experiment should be repeated.

We understand the Reviewer's concern, however, the MDS clearly shows a significant difference in the microbiome between the 20°C and 32°C treatments, which corresponded with significant changes in the DMS(P)(O) data. We agree with the Reviewer that the link between the shift in microbiome and DMS(P)(O) concentrations cannot be directly established in this study, and the speculative aspects of the discussion regarding these potential links have now been scaled-back (**see lines 504-505, 540-542, 547-551**). It is now acknowledged that "These shifts in microbiome structure are likely to have been driven by either the changing physiological state of *A. minutum* cells, shifts in biogenic sulfur concentrations, the presence of other solutes, or a combination of all." (**lines 41-43**).

Also, the bacterial populations would respond to dissolved materials released from the phytoplankton, but there were no measurements aimed at quantifying those releases, making interpretations difficult.

The Reviewer makes a fair point that the microbiome composition will be dictated by a range of biochemical factors, and as stated above we now acknowledge this point on **lines 41-43, 504-505, 540-542 and 547-551.** However, without performing a full metabolomic analysis of the samples, which was beyond the scope and focus of this study, it is not possible to make a priori assessments of the range of chemicals to monitor. Given that *A. minutum* is a prolific DMSP producer, and it is widely hypothesized that DMSP is a key currency in the chemical exchanges between phytoplankton and bacteria, we focused on the role of Sulphur compounds.

Overall, I feel that the manuscript does not make a substantial contribution as it is, primarily because of the extreme temperature used to produce effects.

The use of laboratory conditions to exactly mimic environmental processes is typically highly challenging from a number of perspectives, and accommodations for environment – laboratory variability often need to be made.  Our main goal here was to examine how the heat-stress response of *A. minutum* was reflected in changes in biogenic sulphur cycling and interactions with the microbiome. The temperature range used here was based on substantial pilot studies (described above) that revealed the large shift in temperature that was required to invoke a physiological stress response in the dinoflagellate species in question.  Without increasing the temperature to this level we did not observe a marked physiological response in the dinoflagellate. We now more clearly point out the reasons for the choice of temperature used in the study (see **lines 141-146**) and feel that our observations provide valuable new insights into how the stress response of dinoflagellates can influence biogenic sulphur cycling in coastal habitats.

**Specific comments:**

Title. They really didn't study sulfur cycling so I suggest changing the wording.
In response to the authors concerns we have now changed the title to "Shifts in dimethylated sulfur concentrations and microbiome composition in the red-tide causing dinoflagellate Alexandrium minutum during a simulated marine heat wave."

In Figure 4, the DMSP per cell (0.5 to 1.6 pmol per cell) for Alexandrium minutum is much lower than you report in Introduction for A. minutum (14.2 pmol/cell; line 68). Is there an explanation for that?
We thank the Reviewer for this comment and have added a whole new paragraph **on lines 395-406** to discuss this point. This paragraph now clearly states that : "... DMSP concentrations reported in this study were a degree of magnitude lower ($0.42 \pm 0.04$ to $1.63 \pm 1.70$ pmol cell$^{-1}$) than that previously reported for *A. minutum* (14.2 pmol cell$^{-1}$; Caruana and Malin, 2014;Jean et al., 2005). This is potentially because this culture of *A. minutum* had been isolated from free-living *A. minutum* for a long time (1988) or because culturing conditions failed to mimic the natural biochemical conditions in which this strain of *A. minutum* usually grow. This biochemical difference could potentially reflect that this strain of *A. minutum* in culture is more robust than free-living dinoflagellates of the same species, thereby potentially justifying the need of a 12°C increase in temperature to induce thermal-stress."

L90. When mentioning the 2016 Marine Heat Waves associated with El Nino, give some indication of the temperature increases that occurred.
We have now added this information: "The 2016 MHW that was associated with El Niño Southern Oscillations resulted in an 8°C increase in sea surface temperature leading to the mass coral bleaching of more than 90% of the Great Barrier Reef (Hughes et al., 2017)" (**see line 89-91**).

L131. Julabo, country??
We thank the Reviewer for noticing this omission and have added information has follows: "Temperature and light control was achieved using circulating water heaters (Julabo, USA) and programmable LED lights (Hydra FiftyTwo, AquaIllumination, USA)" (**see lines 133-136**).

L178. 10 ul of H2O2. Give the concentration of H2O2 added and the final concentration in the sample.
We thank the Reviewer for picking that up and have added information as follows: "A positive (+ 10 µL of 30% H2O2, final concentration 97mM) and negative (no ROS added) control of PBS were run to ensure that detected cell fluorescence was completely attributable to the ROS probe." (**line 187**).

L185. The DMS samples were unfiltered. Were they purged for analysis or did you do static headspace? The static headspace would have a relatively high detection limit.
Please provide that value.
Due to the very high DMS concentrations in the Alexandrium cultures, it was possible to analyse DMS concentrations using simple headspace injections as indicated in the methods. The detection limit is now provided as follows: "Detection limit was 50 nM for 500µL headspace injections" (**lines 212-213**).

L188. From the description, the "DMSP" samples would include DMS that was already in the sample. Was this subtracted from the total DMS after the NaOH?
The reviewer makes a good point and we have now corrected our DMSP values to account for the presence of DMS, and have included this extra detail in the methods (**lines 213-215**). Furthermore, Figure 4 and result section (**lines 318-320**) have been amended accordingly. It is important to note that because DMS values corresponded to less than 5% of DMSP values, this amendment did not lead to any substantial change.

L192. The transition here to "after the experiment DMSP samples were opened:" is awkward because they didn't describe yet how the DMSP samples were measured.
They did this by headspace analysis, which is described further down. I suggest reorganizing to make it clearer.
The Reviewer makes a good point and we have now reorganized this whole paragraph accordingly (**see lines 192-229**).

It should be mentioned in methods that all the sulfur compounds were normalized to cell number.
We thank the Reviewer for pointing out this omission and this details has now been added as follows: "All dimethylated sulfur compounds were normalised to cell density, which best reflects biogenic production." (**lines 228-229**)

But normalizing these parameters to the cells may be misleading. While most of the DMSP will be in the cells, the DMS is most certainly not in the cells. The DMSO has an unknown dissolved and particulate partitioning in their cultures. Referring to them as "cellular" concentrations is not correct.
We agree with the Reviewer that DMS and DMSO concentrations should not be referred to as "cellular" since they are most likely not contained within the algal cells. We have thus modified this accordingly throughout the manuscript. However, normalising DMS(P)(O) concentrations to cell numbers remains the most accurate and realistic way to normalise these biogenic sulfur compounds as expressing them in nM without taking algal growth into account would lead to an overestimation of their net production (**see lines 228-229**). It is to be noted that DMS and DMSO are commonly expressed per cell (Hatton and Wilson, 2007; Steinke et al., 2011) or per Chl a concentration (Harada et al., 2009; Bucciarelli et al. 2013) in the literature, which is a very similar approach.

L225. The description of which samples were sequenced is a little vague. They sayvthey sequenced the three highest DNA samples from each treatment at time zero (so 6 samples) and at T=120 h (6 samples). So, a total of 12 samples were sequenced. Isvthat correct? By choosing the three samples with the highest DNA could that bias thevresults?
Yes, the Reviewer's interpretation is correct. By using this approach, we had 6 samples at time 0 (all confirmed to have very similar microbial composition), and 6 samples at time 120 (3 from the 24°C and 3 from the 32°C treatment). Samples with the highest DNA quantity (for which DNA extraction was the most successful) were chosen to ensure cost-effective and successful sequencing. However, this approach should not lead to any inherent bias, as the relative abundance of associated microbes should be similar across all replicated samples from the same treatment, regardless of the DNA concentrations. It is also to be noted that the sequence provider normalises the samples according to the DNA concentrations to ensure sufficient reads from all samples.

They filtered 400 ml onto a 0.22 _m filter, so this would capture both prokaryote and eukaryote DNA. Any interference from all the phytoplankton DNA? They mention removing the chloroplast DNA sequences later on. If the focus here is only the bacteria then the description should be clarified.
We used a bacterial specific 16S rRNA primer set, which will specifically target bacterial DNA, so there should be little influence of the eukaryotic DNA in our sequencing results. Chloroplast sequences were indeed removed, further limiting any influence of the eukaryotic DNA.

L248. I am not an expert in statistics so I can't comment on the approaches used here.
But I will say that it wasn't clear to me whether the relative abundance of bacterial groups in each independent replicate was averaged to obtain an error term.
We were not entirely sure of what the Reviewer was asking here, but a two-way PERMANOVA with Bray-Curtis takes the response variable of each replicate and the error term is derived from the full data set.

L287 Add word : : :compared to the 20_C CONTROL at all time points: : :
We have now amended this text in-line with the Reviewer's comments (**see line 301**)

L 289. You say the 32 deg cultures increased to close to those of the control, but were they still significantly lower?
The Reviewer is correct and we have now clarified this detail in the manuscript, where we state: "However, on days 5 and 6, the FV/FM of cultures kept at 32°C recovered to values (0.72 ± 0.008) close to those of the control (0.75 ± 0.004) (Fig. 2B), although it remained significantly lower than at 20°C (p < 0.01 and p < 0.001 on day 5 and 6, respectively." (lines 302-305)

L396. It should be a negative correlation, not positive.
We thank the Reviewer for noting this typo. We have now changed this text on **line 425-426**.

L436. The statement that algal DMSP lyases seem to be exclusively extracellular, is not correct. The Stefels paper is the only one that reported extracellular lyase activity, and that study might have methodological issues that led to that conclusion. Evidence against extracellular lyase in Phaeocystis (the same genus studied by Stefels) was presented in del Valle et al (2011, Marine Chemistry, 124: 57-67). Admittedly, few studies have looked at this directly, but even from the bacterial side, most of the evidence from natural water samples (algae and bacteria present) points to intracellular degradation of DMSP. This is based on the fact that an inhibitor of DMSP uptake (e.g. glycine betaine), which does not inhibit DMSP lyases, is nearly 100% effective at blocking DMSP degradation (e.g. Li et al. 2016, Environ. Chem. 13: 266). If extracellular lyases were important, DMSP degradation would not be blocked by glycine betaine. Furthermore, the bacterial taxa that were identified to have an extracellular lyase (Alcaligenes sp), and its lyase type (dddY), are not prevalent in marine systems (Moran et al Ann Rev Marine Sci, 2012, 4: 523).
We thank the Reviewer for his/her comment. We have now modified this paragraph accordingly: "Although sporadic, the increases in DMS and DMSO observed in the 32°C treatment may have resulted from enhanced intracellular DMSP cleavage by phytoplankton (Del Valle et al., 2011) or enhanced DMSP exudation from phytoplankton cells during cell lysis (Simó, 2001), resulting in an increasing pool of dissolved DMSP made readily available to both bacteria and phytoplankton DMSP-lyases (Riedel et al., 2015;Alcolombri et al., 2015;Todd et al., 2009;Todd et al., 2007)...” (**see lines 459-470**).

L534. In this conclusion section the authors need to make it clear that the effect was with the extreme 12-degree upshift.
We thank the Reviewer for this suggestion and this change has now been made (**see line 559-564**): “Here, we hypothesized that a very acute increase in temperature, mimicking extreme coastal MHWs, would trigger both a physiological and biochemical stress response in the DMSP-producing dinoflagellate *A. minutum*. This response was indeed observed following a 12°C-increase in temperature, with evidence for impaired photosynthetic efficiency, oxidative stress, spikes in DMS and DMSO concentrations, a drop in DMSP concentration and a shift in the composition of the *A. minutum* microbiome.”

Figures 1 and 2. If you are going to connect the data points as a time trend, you should plot them on a linear x-axis rather than a categorical axis, as presently done. The categorical axis gives a misleading impression of the time trend.
We thank the Reviewer for this suggestion and have modified the x-axis throughout Figures 1, 2, 4 and 5.

Figure 3. The x scale is screwed up. Fv/Fm should be less than 1. It seems they have multiplied it by 100. Please fix.
We thank the Reviewer for noting this. We have now amended Figure 3 accordingly.
The manuscript reports an experiment where a cultured strain of the dinoflagellate Alexandrium minutum was exposed to temperature increases of 4_C and 12_C. Growth rate, photosynthetic efficiency, oxidative stress, dimethylated sulfur compounds and bacterial community composition were measured over several days. The objective of the experiment was to study if an expected decline in growth rate resulting from impaired physiology was accompanied by up-regulated levels of dimethylated sulfur compounds, and if this matched changes in the microbiome that could be related to sulfur-utilizing bacteria. The environmental context for the lab work is the effects of marine heat waves on coastal ecosystems, including harmful algal blooms.

Even though the idea behind the experiments is timely and interesting, the experimental conditions chosen generate a little concern, and the actual results are only partially convincing. Perhaps the authors can provide further convincing arguments with the data at hand.

I will give my comments following the order of the manuscript:

L55: The role of DMSP as a grazing deterrent is, at the least, debatable. It is true that the works of Wolfe et al. and Strom et al. suggested deterrence, but more recent work by one of the authors and others (Seymour et al.) indicated DMSP may be more an attractant than a deterrent.

The Reviewer makes a fair point and in fact, it is the cleavage of DMSP to DMS and acrylate that is believed to have strong deterrent properties for grazers, most likely through the presence of acrylate at high concentrations. We have now changed this sentence to read: "Many marine phytoplankton produce the organic sulfur dimethyl sulfoniopropionate (DMSP) (Zhou et al., 2009;Berdalet et al., 2011;Caruana and Malin, 2014), for which it can function as an antioxidant, osmolyte, chemoattractant and currency in reciprocal chemical exchanges with heterotrophic bacteria (Stefels, 2000;Sunda et al., 2002; Kiene et al., 2000;Seymour et al., 2010)." (lines 52-56)

L80: acute temperature increases – should you say also "ephemeral"?
The Reviewer makes a fair point and we have now made this change (see lines 79 and 387).

L343-349: I do not like the use of the word "driven" here. Should it be "aligned"? What the MDS analysis shows is that, in the 32_C treatment, differences in the microbiome we aligned with elevated ROS, but that the latter drove the former is just a hypothesis.
The same applies to the microbiome composition and abundances in the control, and to the subsequent comparison of variables.
We agree with the Reviewer's comments and have amended this term accordingly throughout the Results section (**see lines 357-364**).

L374: In the case of the San Francisco Bay, MHW were characterized by "increases in temperature of about 8_C above the yearly average". Was it +8_C of the yearly (annual?) average or of the monthly climatological temperatures? +8_C above the annual average would not be too impressive.
The 8°C increase in temperature referred to here was indeed above the monthly average, whereby the MHW occurred during September, with surface water temperatures reaching 22.6°C, while the average temperature for this month is ~ 14°C. We have now clarified this statement on **lines 389-390**, where we now state: "Large increases in temperature of about 8°C above the monthly climatological average led to red-tides of exceptional density in San Francisco Bay (Cloern et al., 2005)".

I mention this because one of my concerns is with the experimental conditions chosen. +12_C seems quite a dramatic treatment. Is there a record of MHW in the S Australian coast where the strain was isolated from? Or perhaps this is not relevant – in any case, what are the temperature shift records of MHW in Australian coasts and elsewhere? More 20_C to 24_C, or 20_C to 32_C?

We agree with the Reviewer and in fact, the next sentence of this paragraph acknowledges this point: "While a 12°C increase in temperature constitutes an extreme scenario of MHWs, even for coastal habitats, this experimental temperature was selected with the intention to induce thermal stress in *A minutum.*"

The amplitude of the temperature increase was dictated by preliminary experiments conducted at 20°C, 24°C, 28°C, 30°C and 32°C, with only a 12°C increase in temperature (32°C) leading to a physiological stress response in this strain of *Alexandrium* in culture. Although an increase in temperature of this magnitude might be rare in coastal marine systems (which is now acknowledged **on lines 143-146**), this presented an opportunity to investigate the biochemical and microbial consequences of thermal stress on this relevant phytoplankton in the context of MHWs. It is also to be noted that cultured *A. minutum* could be more robust than their free living relatives, and in fact they present biochemical differences that are now acknowledged in the manuscript (**see lines 395-407**).

L396: The correlation is negative, not "positive".

We thank the Reviewer for noting this typo. We have changed this **on line 425-426.**

L421-426: I may understand, as a working hypothesis, that optimal growth (hence less physiological stress) could be associated with lower DMS/P/O concentrations per cell.

But it is harder to understand that sulfur concentrations (per culture volume) decreased during the experiment, even with A. minutum being in exponential growth.

The Reviewer is correct and we believe that this interpretation is due to our initially unclear description of the data. What we meant was that the DMS(O) concentrations were significantly lower than in the 20°C control, rather than that the concentrations decreased. We now clarify this point on **lines 449-454**, where we now state: "This temperature optimum was associated with lower DMS and DMSO concentrations than in the 20°C control, although this was only evident 24h after the start of the experiment. Since algal stress responses often result in increased cellular sulfur concentrations in dinoflagellates (McLenon and DiTullio, 2012;Berdalet et al., 2011), it is perhaps not surprising that DMS and DMSO concentrations were lower under what appear to have been more optimal growth temperature conditions."

L434-438: Why do you say that algal DMSP lyases are exclusively located extracellularly? This is definitely not the case in, e.g., Emiliania huxleyi (works by Steinke, Wolfe, Alcolombri).

The Reviewer is correct and we have now modified our text to reflect this, where we now state on **lines 459-464**: "Although sporadic, the increases in DMS and DMSO observed in the 32°C treatment may have resulted from enhanced intracellular DMSP cleavage by phytoplankton (Del Valle et al., 2011) or enhanced DMSP exudation from phytoplankton cells during cell lysis (Simó, 2001), resulting in an increasing pool of dissolved DMSP made readily available to both bacteria and phytoplankton DMSP-lyases (Riedel et al., 2015;Alcolombri et al., 2015;Todd et al., 2009;Todd et al., 2007)."

L446-451: There always is a difficulty when trying to explain and provide experimental evidence for the role of DMS in scavenging ROS: what is first, the decline in DMS or the decline in ROS? It is probably a matter of time scales and potential upregulation by metabolic synthesis. The arguments you provide here carry some assumption that must be explicated.

The Reviewer makes a good point and we are now acknowledging the level of uncertainties in this paragraph by saying: "In contrast, 24h after the start of the experiment, increased ROS coincided with an abrupt decline in DMS and DMSO, perhaps suggestive of serial oxidation via active ROS scavenging of both DMS to DMSO and DMSO to methane sulfinic acid (MSNA) (Sunda et al., 2002), although it is always difficult to confidently link DMS(O) and ROS dynamics unless using tracing techniques." (**see lines 475-479**).

L492: I would replace DMSP metabolism with DMSP catabolism.

The Reviewer makes a fair point and we have now amended this terminology in line with their suggestion (**see line 515**).

The bacterial community composition characterization was not very informative or illustrative with respect to the cycling of sulfur compounds. Very few of the OTUs that increased their abundances under warming had relatives with genes for sulfur compound transformations. I do not find it any surprising – I think it was too naïve to expect that the bacterial community associated with stressed algae relies mainly on sulfur compounds. Instead, I would expect e.g. opportunistic bacteria. So, I agree with what you say in L513-515. However, I do not agree with your statement in L509-512, at least with the wording used. Quick conversion of DMSP to DMS and oxidation of DMS to DMSO is not a reflection of preferential growth of sulfur-consuming bacteria. Actually, DMSP-to-DMS and DMS-to-DMSO are two processes that do not consume sulfur; if anything, they consume carbon or provide energy. Demethylation of DMSP does lead to sulfur consumption and utilization, and this is a competing process to DMSP cleavage.

The Reviewer is correct and we have reworded this section to clarify our point, which we agree was unclear (**see lines 549-559**). What we meant by "…quick conversion of DMSP to DMS and oxidation of DMS to DMSO…" was that "…rapid changes in DMS and DMSO concentrations were potentially caused by (or led to) a shift in microbiome composition towards the preferential growth of sulfur-consuming bacteria." It is now acknowledged that: "the change in microbial abundance could have also been triggered by a range of other parameters that were not measured in this study." (**Lines 541-543**)

Also, you should not base your explanation of the dynamics of the sulfur compounds on the bacterial community alone. There is a potential large role of the dinoflagellate itself: arrest of methionine synthase activity under growth arrest, DMSP cleavage to DMS by the algal lyases, etc.

We agree with the Reviewer and have now included discussion of these potential processes by stating:

"Although sporadic, the increases in DMS and DMSO observed in the 32°C treatment may have resulted from enhanced intracellular DMSP cleavage by phytoplankton (Del Valle et al., 2011) or enhanced DMSP exudation from phytoplankton cells during cell lysis (Simó, 2001), resulting in an increasing pool of dissolved DMSP made readily available to both bacteria and phytoplankton DMSP-lyases (Riedel et al., 2015;Alcolombri et al., 2015;Todd et al., 2009;Todd et al., 2007)." (**lines 459-464**)

"Since this decrease in DMSP at 96h was not coupled with an increase in DMS, this could alternatively be indicative of a decrease in methionine synthase activity (McLenon and DiTullio, 2012) or assimilation of DMSP-sulfur by bacterioplankton for *de novo* protein synthesis (Kiene et al., 2000), with this demethylation pathway often accounting for more than 80% of DMSP turnover in marine surface waters." (**lines 466-470**)

From the figures: The (opposite) patterns of ROS and FvFm are pretty consistent.
Conversely, the patterns of sulfur compounds are less convincing. The fact that the two controls (20_C) show remarkable differences makes one wonder what would have been the results from repeated perturbations. You may need an extra effort to persuade the readers/reviewers of the robustness of the observed responses with respect to the sulfur compounds.

As described in the method, both experiments were conducted at different times and it was thus not to be excluded that the 2 controls kept at 20'C could present some physiological (Fig. 1 & 2) and biochemical (Fig. 4) differences, which perhaps reflected inherent heterogeneity in biological systems. However, the significant differences that were observed between temperature treatments in each experiment were clearly driven by the increase in temperature since both temperatures (control and experimental) were tested at the same time, on the same culture, and under the exact same experimental conditions of light and GSe medium in each experiment.
Because the turnover of DMS(P)(O) in biological systems can occur very quickly (Simo et al 2000), measured changes in DMS(O) concentrations can seem to occur sporadically. However, a clear cascading stress response emerged from these results, which is worth reporting and discussing.
We have now acknowledged variability and uncertainties **on lines 495-499**, which reads: "Because the turnover of DMS, DMSP and DMSO in biological systems can occur very quickly (Simo et al 2000),

DMS and DMSO concentrations can change rapidly, which sometimes makes it difficult to clearly establish cause-effect relationships between physiological stress and the biogenic sulfur response."

L531: Only the "very acute" treatment elicited a response.
This section has been amended based on comments from both Reviewers and it now reads as follows: "Here, we hypothesized that a very acute increase in temperature, mimicking extreme coastal MHWs, would trigger both a physiological and biochemical stress response in the DMSP-producing dinoflagellate *A. minutum*."

References: the reference Simó 2001 is repeated.
We thank the Reviewer for picking this up. This has been amended.

Figure 4b: The difference between treatments is essentially one time point.
We agree with the Reviewer, however, the fact that differences in sulfur concentration between treatments rely on rapid changes in DMS(O)(P) concentrations, reflective of a quick turnover of DMS(P)(O) in biological systems (Simo et al 2000) is now better acknowledged and explained in the discussion (see **lines 495-499**)

[revised manuscript text omitted]

---

## Author Response (AR2)

Response to Reviewers comments

We would like to thank the Reviewer for taking the time to carefully read the former reviewers' reports and to comment on our manuscript. This has greatly helped enhancing a few important points that needed consideration. Specific responses to the Reviewer's comments appear in blue below, with line numbers referring to the manuscript with track changes.

I found this paper to be on a very interesting and important topic. The model organism used was wholly appropriate and I can understand why the authors carried out such a study. I must say that upon reading the comprehensive reviews of the two reviewers that their significant questions are valid. Particularly, I am not certain a 12C hike in temperature is environmentally relevant. However the authors take on this point and acknowledge this, but, I do feel that this could be stated more often and more prominently.

We agree with the Reviewer's comment and have now stated more prominently that a 12°C increase in temperature constitutes an extreme case of MHWs, which has, to our knowledge, not yet been reported in the environment (see lines 45, 109-110, 387, 416-417 and 566). We also state that our temperature treatment was selected, based on preliminary experiments, with the intention to induce thermal stress in this particularly robust strain of *Alexandrium minutum* (lines 143-148).

As with the two reviewers, I agree that the temperature increases are having an effect on the growth and physiology of the dinoflagellate, but I am less than convinced on the effects of the temperature hikes on the cycling of organic sulfur compounds, which is one of the key aims of the study.

-I am afraid that I am not certain of effects of the 24C and 32C treatments on DMS and DMSO standing stocks, since in all cases effects are only seen in one time point (6h for 32C and 24h for 24C). Perhaps the authors are not likely to capture changes in DMSP/DMS/DMSO when only looking at studying standing stocks. This should be acknowledged in the manuscript.

We understand the Reviewer's concern in regards to the concentrations of sulfur compounds and we agree that the differences in DMS and DMSO concentrations over time between the control and high temperature treatments are very subtle. The reviewer makes a good point about the potential limitations of measuring standing stocks, which we now acknowledge in the manuscript where we state "It is also to be noted that measuring standing stocks may constitute a limitation to capture subtle changes in DMS, DMSP and DMSO over time." (lines 483-485).

There is no noticeable change in DMSP standing stocks across the experiments until 96 h in the 33C experiment. Even then it seems to be equalling up at 120h. It might be more convincing if the authors had monitored changes in gene transcription for DMSP synthesis and lysis genes in A. minutum over the experiment. I do appreciate though that this is not easy and I am not asking for this to be done here.

The reviewer again makes an interesting point, but an analysis of gene regulation was unfortunately outside of the scope of this study.

-Also I do not understand why the 20C controls for the 24C and 32C experiments have such different profiles (e.g., Panel E and F of figure 4)? If I am understanding it correctly they should have very similar profiles? If I am understanding it correctly then some of the differences between the two 20c incubations are more dramatic than the differences reported here for the temp hikes, e.g., the DMSO production in panel's e and f of Figure 4. I may have misinterpreted the experiments here. Apologies if I have. Can this be explained?

We believe that the different sulfur profiles observed between the two 20°C treatments are probably a consequence of the experiments being conducted at a different times (April and June, see line 132), whereby changes in the physiological state of the culture at each time led to different levels of DMSO. We now acknowledge this variability and its potential source in the figure caption where we have now added a sentence stating: "Variability in between the two 20°C control is probably a consequence of experiments 1 and 2 being conducted at a different times (April and June), whereby changes in the physiological state of the culture at each time led to different dimethylated sulfur profiles."

-Given the community change work was done at 120h when DMS and DMSO levels are similar to control samples, I feel it is most likely the temperature may be governing the change in microbial community and not the organic sulfur molecules they make? This should be stated in the manuscript.

The Reviewer makes a fair point and we have now indicated that temperature alone could have contributed to the shift in the microbial community on lines 42 and 556, where we state: "These shifts in microbiome structure are likely to have been driven by either temperature itself, the changing physiological state of A. minutum cells, shifts in biogenic sulfur concentrations, the presence of other solutes, or a combination of all." And "Alternatively, the observed shifts in microbiome structure may have occurred independently to the biogenic sulfur cycling processes and was instead related to either temperature itself or other metabolic shifts in the heat-stressed *A. minitum*."

Generally I feel that the authors of this manuscript have done a good job answering the reviewer's points and making the manuscript more balanced. In conclusion, I feel that the manuscript is worthy of publication here if the above concerns are dealt with.

We thank the Reviewer for his/her insightful comments and suggestions and for recognising the potential of our manuscript.

Extra points to raise:
-on line 43Indictae temperature itself as a potential driver for community change.

We have now made this addition on line 42.

-L52 sulfur "compound"

This has now been added on line 53.

-L75 A. minutum

This has now been corrected (line 76).

-L86-94 can you give an example of a =12C hike in temp?

Unfortunately, we cannot provide a specific example of a 12'C increase in temperature recorded in the environment. However, we acknowledged in the text that although a 12°C increase in temperature constitutes an extreme scenario of MHWs, even for coastal habitats, this experimental temperature was selected after preliminary investigations with the intention to induce thermal stress in this strain of *A minutum* in culture (see lines 393-396).

-L109 Specifically state 12C hike in temp and say if natural or not.

This point has been clarified as follows: "The aims of this study were to investigate how an acute increase in temperature (+12°C), comparable to those associated with MHW events and leading to thermal stress in *A. minutum* could alter the physiological state and biogenic sulfur cycling dynamics of *A. minutum*." (see lines 109-112)

-L223 I may have a problem here with how you assay for DMSO. My problem is that if A. minutum produces DMSOP (Thume et al., Nature. 2018 Nov;563(7731):412-415.) then your assay for DMSO described here will also include DMSO derived from DMSOP that has been chemically cleaved.

We understand the Reviewer's concern about DMSOP potentially being a source of DMSO in this study, and thus altering the amount of DMSO derived from DMSP and DMS oxidation. However, the presence of DMSOP has only recently been discovered in the marine environment (Thume et al, 2018) and there is no evidence for Alexandrium minutum to contain/produce this sulfur compound. Furthermore, it is also acknowledged in Thumes et al (2018) that DMSO is mainly produced via DMS oxidation.

L516 You have the wrong reference for dsyB. This should be Curson et al., Nat Microbiol. 2017 Feb 13;2:17009.

We apologise for this error. This has now been corrected with the suitable reference added.

L560 Mimicking extreme coastal MHW's is this true? Give example.

Since MHWs are defined as an abrupt and ephemeral increase in temperature of at least 3 to 5°C above climatological average that lasts for at least 3 to 5 days (lines 388-389), and since we cannot provide a specific example of a 12°C increase in temperature that occurred in the marine environment, it is true that this experiment mimics an extreme case of coastal MHWs.

-It is perhaps worth mentioning more prominently that A. minutum is very robust in relation to temperature changes and that actually it is not likely to be affected by environmentally relevant temperature hikes. This almost definitely will be a strain specific phenomenon.

[revised manuscript text omitted]